# Humans use forward thinking to exploit social controllability

**Soojung Na[1,2,3†], Dongil Chung[4†], Andreas Hula[5], Ofer Perl[3], Jennifer Jung[6], Matthew Heflin[3], Sylvia Blackmore[3,7], Vincenzo G Fiore[3], Peter Dayan[8,9], Xiaosi Gu[2,3]***

[1]The Graduate School of Biomedical Sciences, Icahn School of Medicine at Mount Sinai, New York, United States; [2]Nash Family Department of Neuroscience, Icahn School of Medicine at Mount Sinai, New York, United States; [3]Department of Psychiatry, Icahn School of Medicine at Mount Sinai, New York, United States; [4]Department of Biomedical Engineering, Ulsan National Institute of Science and Technology, Ulsan, Republic of Korea; [5]Austrian Institute of Technology, Seibersdorf, Austria; [6]School of Behavioral and Brain Sciences, The University of Texas at Dallas, Richardson, United States; [7]Queen Square Institute of Neurology, University College London, London, United Kingdom; [8]Max Planck Institute for Biological Cybernetics, Tübingen, Germany; [9]University of Tübingen, Tübingen, Germany

**Abstract** The controllability of our social environment has a profound impact on our behavior and mental health. Nevertheless, neurocomputational mechanisms underlying social controllability remain elusive. Here, 48 participants performed a task where their current choices either did (Controllable), or did not (Uncontrollable), influence partners' future proposals. Computational modeling revealed that people engaged a mental model of forward thinking (FT; i.e., calculating the downstream effects of current actions) to estimate social controllability in both Controllable and Uncontrollable conditions. A large-scale online replication study (n=1342) supported this finding. Using functional magnetic resonance imaging (n=48), we further demonstrated that the ventromedial prefrontal cortex (vmPFC) computed the projected total values of current actions during forward planning, supporting the neural realization of the forward-thinking model. These findings demonstrate that humans use vmPFC-dependent FT to estimate and exploit social controllability, expanding the role of this neurocomputational mechanism beyond spatial and cognitive contexts.

**\*For correspondence:**
xiaosi.gu@mssm.edu

[†]These authors contributed equally to this work

**Competing interest:** The authors declare that no competing interests exist.

## Introduction

Humans do not always have influence over the environments which they occupy. A lack of controllability has a profound impact on mental health, as has been demonstrated by decades of research on uncontrollable stress, pain, and learned helplessness (*Maier and Seligman, 1976*; *Maier and Watkins, 2005*; *Overmier, 1968*; *Weiss, 1968*). Conversely, high levels of controllability have been associated with better mental health outcomes such as higher subjective well-being (*Lachman and Weaver, 1998*) and less negative affect (*Maier and Seligman, 2016*; *Southwick and Southwick, 2018*). For humans, one of the most important types of controllability we need to track concerns our social environment. Doing this could be one of the roles of the various neural systems whose involvement in social cognition is supported by mounting evidence (*Atzil et al., 2018*; *Dunbar and Shultz, 2007*). Nevertheless, despite the importance, the neurocomputational mechanisms underlying social controllability have not been systematically investigated.

Based on previous work demonstrating the computational mechanisms of controllability in non-social environments, here we hypothesize that people use mental models to exploit social controllability,

for instance via forward simulation. In non-social contexts, it has been proposed that controllability quantifies the extent to which the acquisition of outcomes, and particularly desired outcomes, can be influenced by the choice of actions (*Huys and Dayan, 2009*; *Dorfman and Gershman, 2019*; *Ligneul, 2021*). In these non-social settings, agents need to learn the association between actions and state (event) transitions and potential outcomes in order to simulate future possibilities (*Pezzulo et al., 2013*; *Szpunar et al., 2014*) and make decisions (*Daw et al., 2011*; *Dolan and Dayan, 2013*; *Doll et al., 2015*; *Gläscher et al., 2010*). It has also been hypothesized that both under- and over-estimation of controllability could be detrimental to behavior (*Huys and Dayan, 2009*) depending on the complexity of the environment. Yet, it remains unknown whether this is true for social controllability.

Studies on strategic decision-making (*Camerer, 2011*) have provided initial insight into the possible mechanisms underlying social controllability and influence. For example, *Hampton et al., 2008* showed that people can learn the influence of their own actions on others during an iterative inspection game; and that the medial prefrontal cortex (mPFC) tracked expected reward given the degree of expected influence (*Hampton et al., 2008*). In other types of strategic games such as bargaining, it has been suggested that individuals differ drastically in their ability to manage their social images and exert influence on others, a behavioral phenomenon subserved by underlying neural differences in prefrontal regions (*Bhatt et al., 2010*). Furthermore, through the application of an interactive partially observable Markov decision process model, *Hula et al., 2015* found that humans are able to use forward planning and mentally simulate future interactions in an iterative trust game (*Hula et al., 2015*). All of these studies suggest that learning the structure of the social environment is crucial for exerting influence, yet none have systematically examined the computational underpinnings of social controllability in a group setting where an agent plays with multiple other players that constitute a more social-like environment.

Neurally, along with recent findings about its role in providing a representational substrate for cognitive tasks (*Behrens et al., 2018*; *Niv, 2019*; *Schuck et al., 2016*), the ventromedial prefrontal cortex (vmPFC) has been shown to signal expected values across a wide range of settings (*Boorman et al., 2009*; *Kable and Glimcher, 2007*; *FitzGerald et al., 2009*; *Behrens et al., 2008*; *Bartra et al., 2013*; *Venkatraman et al., 2009*). The majority of studies have focused on the role of the vmPFC in encoding the subjective values of non-social choices (*Boorman et al., 2009*; *FitzGerald et al., 2009*; *Kable and Glimcher, 2007*; *Venkatraman et al., 2009*). Nevertheless, accumulating evidence also pinpoints to a central role of the vmPFC in computing the value of social choices (*Behrens et al., 2008*; *Hampton et al., 2008*; *Hiser and Koenigs, 2018*), such as expected values computed based on learned influence (*Hampton et al., 2008*). A recent meta-analysis suggests that both social and non-social subjective values reliably activate the vmPFC (*Bartra et al., 2013*). Thus, we expect that the vmPFC will also play an important role in social controllability where the value of future events should be simulated and computed.

In the current study, we hypothesize that humans exploit social controllability by implementing forward thinking (FT) and mentally simulating future interactions. In particular, we consider the long-lasting effect that one's current interaction with one other person can have on future interactions with many others who constitute the social environment, for instance by developing a reputation. We predict that social agents will use forward planning to take into account not only decision variables related to the present interaction with a current partner, but also those related to future interactions with other partners from the same milieu. Finally, we hypothesize that the choice values integrating the planned paths would be signaled in the vmPFC.

We used computational modeling and functional magnetic resonance imaging (fMRI; n=48), in the context of a social exchange paradigm (see *Figure 1* and Materials and methods), to test the hypothesis that FT serves as a mechanism for social controllability. Furthermore, we replicated our computational findings in a large-scale online study involving more geographically diverse participants (n=1342). Both in-person and online participants completed an economic exchange task where they did (Controllable) or did not (Uncontrollable) influence their partners' proposals of monetary offers in the future (see *Figure 1a and b*, and Materials and methods for details). Participants were told that they were playing members coming from two different teams, one each for the two controllability conditions (in a counterbalanced order across subjects); in fact, they played with a computer algorithm in both cases. *Supplementary file 2* provides the task instruction provided to participants. To directly compare the impact of social versus non-social contexts on individuals' decision strategies,

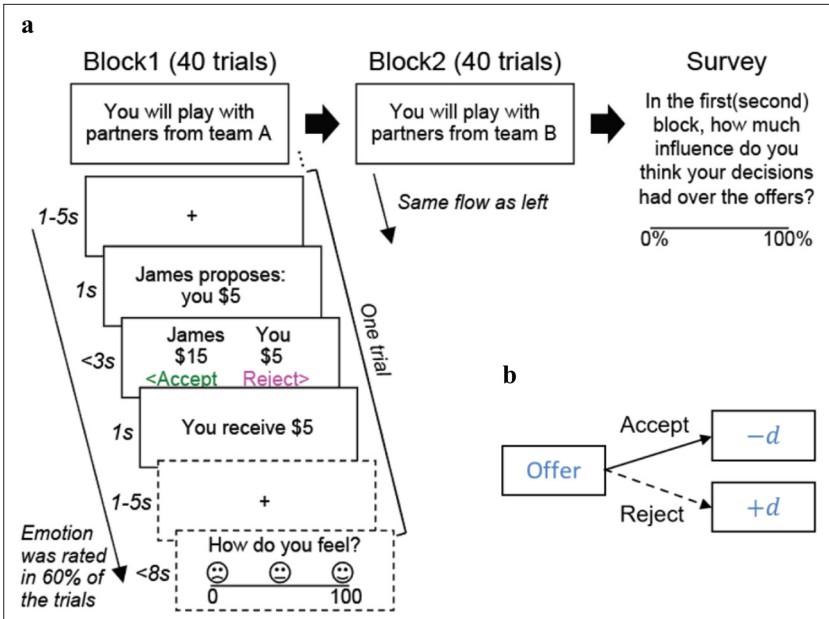

**Figure 1.** Experimental paradigm. (**a**) Participants played a social exchange task based on the ultimatum game. There were two blocks: one 'Controllable' condition and one 'Uncontrollable' condition. Order of the conditions was counterbalanced across participants. Each block had 40 (fMRI sample) or 30 (online sample) trials. In each trial, participants needed to decide whether to accept or reject the split of $20 proposed by virtual members of a team. In the fMRI study, participants rated their emotions after their choice in 60% of the trials. Upon the completion of the game, participants rated their subjective beliefs about controllability for each block. (**b**) The schematic of the offers (the proposed participants' portion of the split) generation under the Controllable condition. Under the Controllable condition, if participants accepted the offer at trial t, the next offer at trial t+1 decreased by d={0, 1, or 2} (1/3 chance each). If they rejected the offer, the next offer increased by d={0, 1, or 2} (1/3 chance for each option). Such contingency did not exist in the Uncontrollable condition where the offers were randomly drawn from a Gaussian distribution (μ=5, σ=1.2, rounded to the nearest integer, max=8, min=2) and participants' behaviors had no influence on the future offers.

The online version of this article includes the following figure supplement(s) for figure 1:

**Figure supplement 1.** Emotion ratings.

we further administered a matched controllability experiment where participants were explicitly told that they were playing against a computer algorithm (*Figure 2—figure supplement 1* and *Supplementary file 1a*).

Participants played against each team as the responder in a social exchange game adapted from the ultimatum game (*Camerer, 2011*) (single-shot games with 40 different partners (rounds) per team for the fMRI sample, and 30 rounds for the online sample). In the Uncontrollable condition, on each round, participants were offered a split of $20 from their partners and asked to decide whether to accept or reject the offer. Unbeknownst to participants, the actual offer was randomly drawn from a normal distribution (rounded and restricted to be between $2 and $8 (inclusive) for the fMRI sample and between $1 and $9 (inclusive) for the online sample; the first offer was always $5). Here, participants' current choices had no influence on the next offers from their partners. The Controllable condition was the same except that participants could exert control over their partners using their own actions. Specifically, participants' current decisions (i.e., to accept or reject the offer) influenced the next offers from their partners in a systematic manner. Subject only to being between $1 and $9 (inclusive), partners increased the next offer by $0, $1, or $2 (probability of ⅓ each, subject to the constraints) if the participant rejected the present offer, and decreased the next offers by $0, $1, or $2 (probability of ⅓ each, again subject to the constraints) if the participant accepted the current offer (*Figure 1b* and Materials and methods). Again, the starting offer was $5. At the end of the task, after all the trials were completed, we asked participants to rate how much control they believed they had over their partners' offers in each condition using a 0–100 scale to measure their perceived action-offer contingency

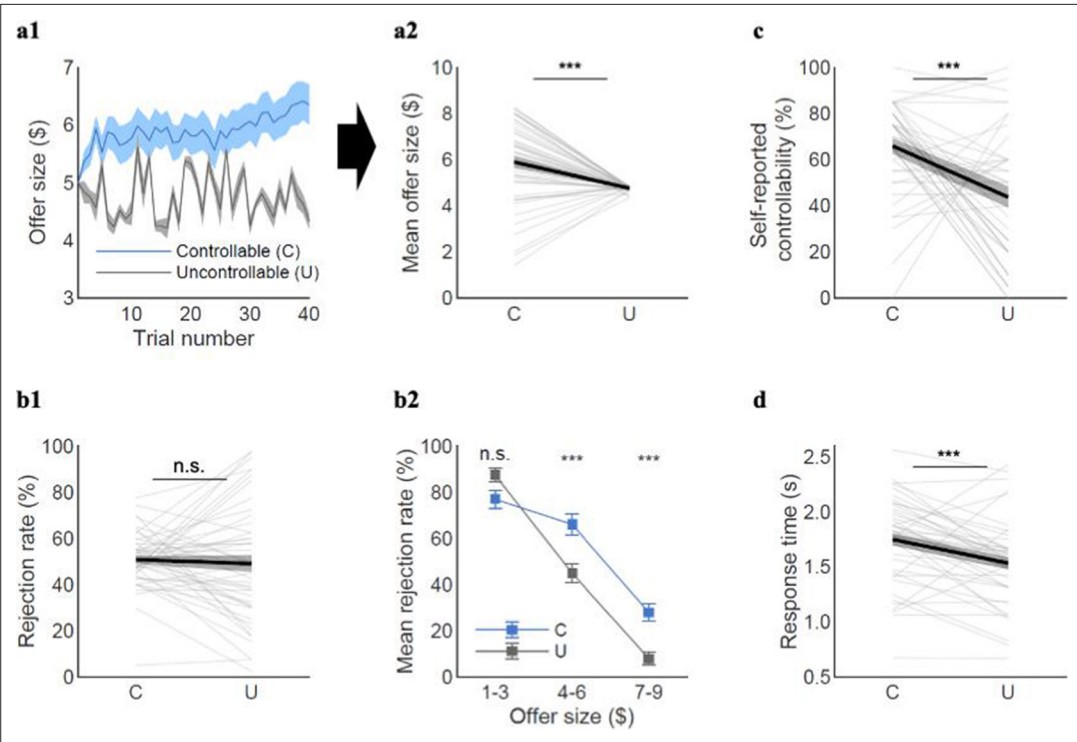

**Figure 2.** Model-agnostic behavioral results. (**a1**) Participants raised the offers along the trials when they had control (Controllable), compared to when they had no control (Uncontrollable). (**a2**) The mean offer size was higher for the Controllable (C) than Uncontrollable (U) condition (mean$_C$=5.9, mean$_U$=4.8, t(47.45)=4.33, p<0.001). (**b1**) Overall rejection rates were not different between the two conditions (mean$_C$=50.8%, mean$_U$=49.1%, t(67.87)=0.43, p=0.67). (**b2**) However, participants were more likely to reject middle and high offers when they had control (low ($1–3): mean$_C$=77%, mean$_U$=87%, t(22)=–1.35, p=0.19; middle ($4–6): mean$_C$=66%, mean$_U$=45%, t(47)=5.41, p<0.001; high ($7–9): mean$_C$=28%, mean$_U$=8%, t(72.50)=4.00, p<0.001). Each offer bin for the Controllable in (**b2**) represents 23, 48, and 41 participants who were proposed the corresponding offers at least once, whereas each bin for the Uncontrollable represents all 48 participants. The t-test for each bin was conducted for those who had the corresponding offers for both conditions. (**c**) The self-reported controllability ratings were higher for the Controllable than Uncontrollable condition (mean$_C$=65.9, mean$_U$=43.7, t(74.55)=4.10, p<0.001; eight participants were excluded due to missing data). (**d**) Response times were longer for the Controllable than the Uncontrollable condition (mean$_C$=1.75±0.38, mean$_U$=1.53±0.38; paired t-test t(47)=4.34, p<0.001), suggesting that participants were likely to engage more deliberation during decision-making in the Controllable condition. A paired t-test was used for the rejection rates for low and middle offers and the self-reported controllability ratings. The t-statistics for the mean offer size, overall rejection rate, rejection rate for high offers, and self-reported controllability are from two-sample t-tests assuming unequal variance using Satterthwaite's approximation according to the results of the F-tests for equal variance. Error bars and shades represent SEM; ***p<0.001; n.s. indicates not significant. For (**a2**, **b1**, **c**, **d**), each line represents a participant and each bold line represents the mean.

The online version of this article includes the following figure supplement(s) for figure 2:

**Figure supplement 1.** Behavioral results of a non-social controllability task.

**Figure supplement 2.** Rejection rates as a function of offer size.

**Figure supplement 3.** Response time.

**Figure supplement 4.** Shift ratio.

('self-reported/perceived controllability' hereafter). In the fMRI study, on 60% of the trials, participants were also asked about their emotional state (How do you feel?) on a scale of 0 (unhappy) to 100 (happy) after they made a choice (i.e., 24 ratings per condition; see *Figure 1—figure supplement 1*).

Note that participants were not instructed about the statistics of the task environment nor the nature of the condition they were playing, although the instruction about the existence of two separate teams was provided to encourage participants to learn contingent rules and norms within each condition (*Supplementary file 2*). If participants were able to detect social controllability correctly

within each condition, they would show strategic decisions that exert appropriate levels of control over others' subsequent choices.

## Results

### Participants distinguished between controllable and uncontrollable environments

We first examined whether participants' choices were sensitive to the difference in controllability between the two social environments, noting that there was no explicit instruction about this difference. Our primary measures here were the offer sizes participants received in each condition, their rejection behavior, and their self-reported controllability. If individuals learned the action-offer contingency of the controllable environment, we should observe that (1) offers received under the Controllable condition would be pushed up to a higher level than those under the Uncontrollable condition; (2) people would need to reject more offers to obtain larger future offers under the Controllable than the Uncontrollable condition; and (3) people would report higher self-reported controllability for the Controllable than for the Uncontrollable condition.

First, we found that despite the same starting offer of $5, participants indeed received higher offers over time under the Controllable compared to the Uncontrollable condition (mean$_C$=5.9, mean$_U$=4.8, t(47.45)=4.33, p<0.001; *Figure 2a1, a2*), indicating that individuals in general successfully exerted influence over the offers made by partners when they were given control.

Next, we examined the rejection patterns from the two conditions. On average, rejection rates in the two conditions were comparable (mean$_C$=50.8%, mean$_U$=49.1%, t(67.87)=0.43, p=0.67; *Figure 2b1*). By separating the trials each individual experienced into three levels of offer sizes (low: $1–3, medium: $4–6, and high: $7–9) and then aggregating across all individuals, we further examined whether rejection rates varied as a function of offer size. We found that participants were more likely to reject medium to high ($4–9) offers in the Controllable condition, while they showed comparable rejection rates for the low offers ($1–3) between the two conditions (low ($1–3): mean$_C$=77%, mean$_U$=87%, t(22)=−1.35, p=0.19; middle ($4–6): mean$_C$=66%, mean$_U$=45%, t(47)=5.41, p<0.001; high ($7–9): mean$_C$=28%, mean$_U$=8%, t(72.50)=4.00, p<0.001; *Figure 2b2*; see *Figure 2—figure supplement 2* for rejection rates by each offer size). These results suggest that participants behaved in a strategic way to utilize their influence over the partners. One possible confound is that individuals may have experienced different affective states in the two conditions and changed their choice behaviors. However, this seemed unlikely because there was no significant difference in emotional rating between the Controllable and the Uncontrollable conditions (*Figure 1—figure supplement 1*).

As additional evidence that participants distinguished the controllability between conditions, we compared self-reported beliefs about controllability between the two conditions. Indeed, participants reported higher self-reported controllability for the Controllable than the Uncontrollable condition (mean$_C$=65.9, mean$_U$=43.7, t(74.55)=4.10, p<0.001; *Figure 2c*). Besides the clear indication of individuals' recognition of the difference in controllability between conditions, the mean level of self-reported controllability for the Uncontrollable condition was 43.7%, which was still substantially higher than their actual level of controllability on future offers made by the partners (0%). This result might suggest that participants could develop an illusory sense of control when they had no actual influence over their partners' offers.

In addition, we examined response times as an exploratory analysis and found that participants took longer time to make their decisions in the Controllable condition than the Uncontrollable condition. These results again suggest that participants differentiated the controllability between conditions (mean$_C$=1.75±0.38, mean$_U$=1.53±0.38; paired t-test t(47)=4.34, p<0.001; *Figure 2d*). Taken together, these findings demonstrate that participants were able to exploit and perceive their influence in a social environment when they had influence, although they have developed an illusion of control, at least to some degree, even when controllability did not exist. We delineate the computational mechanisms underlying these behaviors in the next sections.

### Participants used forward thinking to exploit social controllability

We constructed computational models of participants' choices and sought to investigate what cognitive processes might underlie people's ability to exploit social controllability. Previous studies on

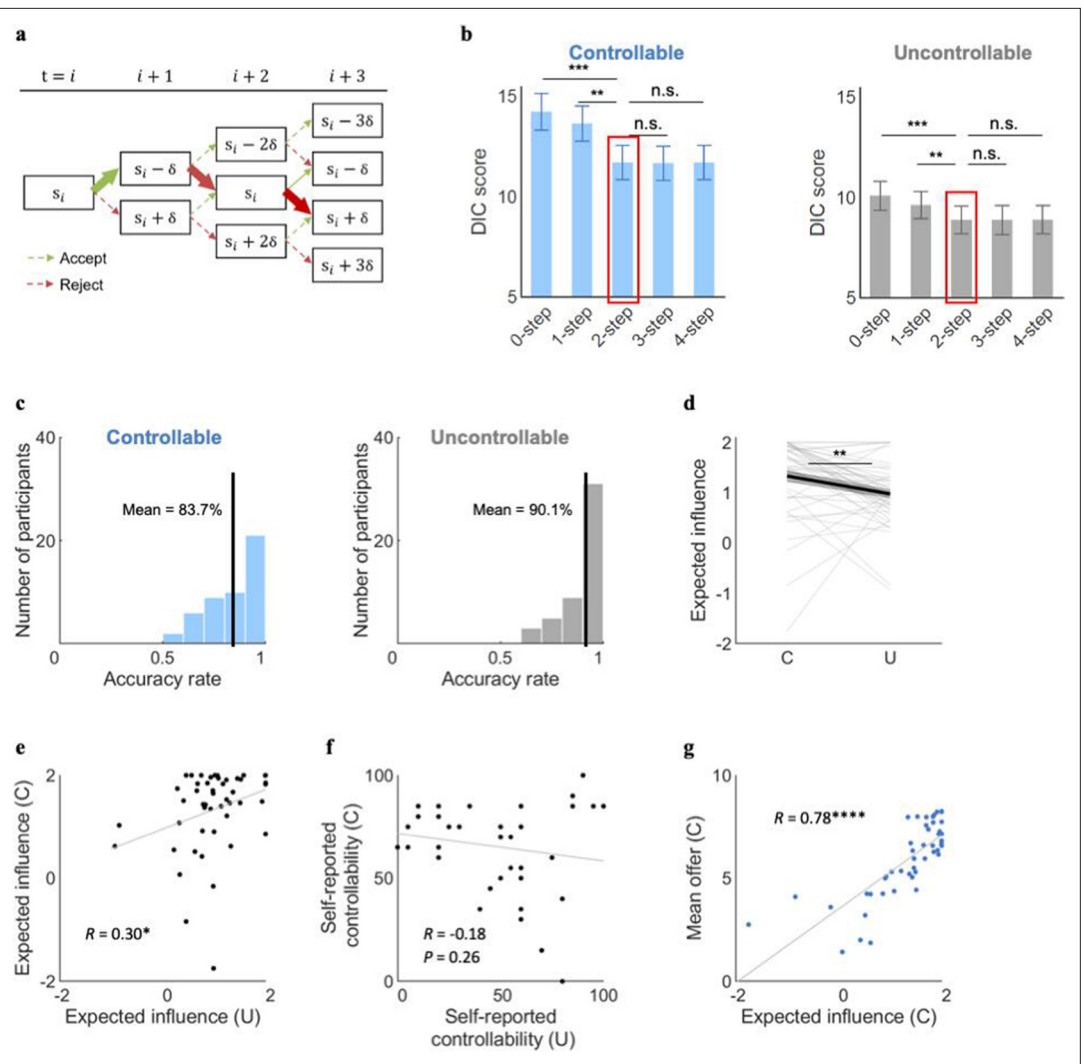

**Figure 3.** Computational modeling of social controllability. (**a**) The figure depicts how individuals' simulated value of the offers evolves contingent upon the choices along the future steps under the Controllable condition. Future simulation was assumed to be deterministic (only one path is simulated instead of all paths being visited in a probabilistic manner). The solid and thicker arrows represent an example of a simulated path. To examine how many steps along the temporal horizon participants might simulate to exert control, we tested the candidate models considering from zero to four steps of the future horizon. (**b**) For both the Controllable and Uncontrollable conditions, the forward thinking (FT) models better explained participants' behavior than the 0-step model. The 2-step FT model was selected for further analyses, because the improvement in the DIC score (Draper's Information Criteria; **Draper, 1995**) was marginal for the models including further simulations (paired t-test comparing 2-step FT model with (i) 0-step Controllable: $t(47)=-4.45$, $p<0.0001$, Uncontrollable: $t(47)=-4.21$, $p<0.001$; (ii) 1-step Controllable: $t(47)=-4.41$, $p<0.0001$, Uncontrollable: $t(47)=-3.01$, $p<0.001$; (iii) 3-step Controllable: $t(47)=0.39$, $p=0.70$, Uncontrollable: $t(47)=-0.04$, $p=0.97$; (iv) 4-step Controllable: $t(47)=0.06$, $p=0.95$, Uncontrollable: $t(47)=-0.12$, $p=0.91$). (**c**) The choices predicted by the 2-step FT model were matched with individuals' actual choices with an average accuracy rate of 83.7% for the Controllable and 90.1% for the Uncontrollable. Each bold black line represents mean accuracy rate. (**d**) The levels of expected influence drawn from the 2-step FT model were higher for the Controllable than the Uncontrollable (mean$_C$=1.33, mean$_U$=0.98, $t(47)=2.90$, $p<0.01$). Each line represents a participant and each bold line represents the mean. (**e**) The expected influence was positively correlated between the Controllable and the Uncontrollable conditions ($R=0.30$, $p<0.05$). (**f**) The self-reported controllability was not significantly correlated between the conditions ($R=-0.18$, $p=0.26$). (**g**) Under the Controllable condition, expected influence correlated with mean offers ($R=0.78$, $p<<0.0001$). Each dot represents a participant. Error bars and shades represent SEM; ****$p<0.0001$; ***$p<0.001$; **$p<0.01$; *$p<0.05$. C, controllable; U, Uncontrollable.

The online version of this article includes the following figure supplement(s) for figure 3:

*Figure 3 continued on next page*

*Figure 3 continued*

**Figure supplement 1.** Model recovery analyses.

**Figure supplement 2.** Adaptive norm learning versus static norm models.

**Figure supplement 3.** Parameter recovery.

**Figure supplement 4.** fMRI sample: results without those who had negative deltas.

**Figure supplement 5.** Comparison with model-free (MF) learning.

value-based decision-making have shown that people can use future-oriented thinking and mentally simulate future scenarios when their current actions have an impact on the future (***Daw et al., 2011***; ***Gläscher et al., 2010***; ***Lee et al., 2014***; ***Moran et al., 2019***). Relying on this framework, we hypothesized that individuals use FT to estimate the impact of their behavior on future social interactions.

To test this hypothesis, we constructed a set of FT models which assume that an agent computes the values of action (here, accepting or rejecting) by summing up the current value (CV) and the future value (FV) based on her estimation of the amount of controllability she has over the social interactions. These models also incorporate social norm adaptation (***Gu et al., 2015***) to characterize how individuals' aversion thresholds to unfairness is adjusted by observing the counterpart teams' proposals (***Fehr and Schmidt, 1999***) (see Materials and methods for details). The key individual-level parameter-of-interest in this model is the 'expected influence,' $\delta$, representing the amount of the offer changes that participants thought they would induce by rejecting the current offer (see Materials and methods). We constrained the range of $\delta$ using a sigmoid function to −\$2 to \$2, in order to match with the range participants observed in the Controllable condition (\$0–2) and to encompass what could happen in the Uncontrollable condition (−\$2 to \$0). Moreover, we considered the number of steps one calculates into the future (i.e., planning horizon; *Figure 3a*). We compared models that considered from one to four steps further in the future in addition to standalone social learning ('0-step;' also see *Figure 3—figure supplement 5* for comparison with a model-free [MF] learning). The 0-step model only considers the utility at the current state. All other components including the utility function of the immediate rewards, and the variable initial norm and norm learning incorporated in the utility function are shared across all the candidate models. In model fitting, we excluded the first 5 out of 40 trials for the fMRI sample (30 trials for the online sample) to exclude initial exploratory behaviors and to focus on stable estimation of controllability. We also excluded the last five trials because subjects might adopt a different strategy toward the end of the interaction (e.g., 'cashing out' instead of trying to raise the offers higher).

The results showed that for both conditions (Controllable, Uncontrollable), all FT models significantly better explained participants' choices than the standalone norm learning model without FT (0-step model) (***Gu et al., 2015***), as indexed by Draper's Information Criteria (DIC) (***Draper, 1995***) scores averaged across individuals (paired t-test comparing 2-step FT model with 0-step model Controllable: t(47)=−4.45, p<0.0001; Uncontrollable: t(47)=−4.21, p<0.001; *Figure 3b*). In addition, not all parameters were recoverable in parameter recovery analysis using the 0-step model (e.g., sensitivity to norm violation; Controllable: r=−0.03, p=0.82; Uncontrollable: r=0.20, p=0.15) whereas all the parameters from the FT models were identifiable (see *Figure 3—figure supplement 3a-j* for

**Table 1.** Parameter estimates from the 2-step forward thinking (FT) model.

| Mean (SD) | Inverse temperature | Sensitivity to norm violation | Initial norm | Adaptation rate | Expected influence |
|---|---|---|---|---|---|
| | $\beta$ | $\alpha$ | f0 | $\varepsilon$ | $\delta$ |
| **Controllable** | | | | | |
| fMRI sample | 8.33 (8.55) | 0.76 (0.29) | 8.21 (7.14) | 0.24 (0.24) | 1.33 (0.79) |
| Online sample | 9.77 (8.54) | 0.74 (0.29) | 9.01 (7.26) | 0.32 (0.31) | 1.34 (0.84) |
| **Uncontrollable** | | | | | |
| fMRI sample | 10.38 (8.84) | 0.79 (0.31) | 8.84 (6.96) | 0.29 (0.24) | 0.98 (0.62) |
| Online sample | 12.94 (7.66) | 0.78 (0.23) | 9.07 (6.31) | 0.24 (0.24) | 0.90 (1.06) |

parameter recovery of the 2-step model). These results suggest that participants engaged in future-oriented thinking and specifically, calculated how their current choice might affect subsequent social interactions, regardless of the actual level of controllability of the environment.

The FT models with longer planning horizon tend to show smaller DIC scores (i.e., better model fit), but the fit improvement became marginal after two steps (paired t-test comparing 2-step FT model with (i) 1-step Controllable: t(47)=–4.41, p<0.0001, Uncontrollable: t(47)=–3.01, p<0.001; (ii) 3-step Controllable: t(47)=0.39, p=0.70, Uncontrollable: t(47)=–0.04, p=0.97; (iii) 4-step Controllable: t(47)=0.06, p=0.95, Uncontrollable: t(47)=–0.12, p=0.91; *Figure 3b*). The 2-step FT model predicted participants' choices with an average accuracy rate of 83.7% for the Controllable and 90.1% for the Uncontrollable condition (*Figure 3c*), which was higher than the 1-step model for the Controllable condition (Controllable 78.4% (t(47)=–3.63, p<0.001), Uncontrollable 88.7% (t(47)=–1.45, p=0.15)) and comparable with the models with longer planning horizon (3-step model: Controllable 84.0% (t(47)=0.20, p=0.84), Uncontrollable 90.7% (t(47)=0.62, p=0.53); 4-step model: Controllable 84.0% (t(47)=0.21, p=0.84), Uncontrollable 90.2% (t(47)=0.09, p=0.93)). Particularly, the parameter of our interest, expected influence $\delta$, was better identified and recovered in general for the 2-step model (Controllable r=0.87, Uncontrollable r=0.79) compared to the other models (1-step model: Controllable r=0.80, Uncontrollable r=0.68; 3-step model: Controllable r=0.81, Uncontrollable r=0.68; 4-step model: Controllable r=0.89, Uncontrollable r=0.68). We thus used parameters from the 2-step FT model for subsequent analyses (see *Table 1* for a full list of parameters from this model).

It might seem counterintuitive that participants engaged a 2-step FT model to estimate the future impact of their current choices under the Uncontrollable condition. However, as in most real-life situations where the controllability of our social interactions is unknown or uncertain, participants were not explicitly told about the uncontrollability of the environment. Indeed, they incorrectly estimated that they could exert at least some control (*Figure 2c*). Thus, we infer that individuals attempted to make strategic decisions with belief that they have some controllability over the social environment independent of the actual controllability.

Given that participants were successful in raising offers in the Controllable condition (*Figure 2a*), we predicted that the expected influence parameter $\delta$ would differ between the two conditions. Indeed, we found that the expected influence parameter estimates drawn from the 2-step FT model were higher for the Controllable than for the Uncontrollable condition (mean$_C$=1.33, mean$_U$=0.98, t(47)=2.90, p<0.01; *Figure 3d*), indicating that participants simulated greater levels of controllability when environments were in fact controllable than when they were uncontrollable. Interestingly, despite the systematic difference between the two conditions, the expected influence was still positively correlated between the conditions (r=0.30, p<0.05; *Figure 3e*), suggesting a trait-like characteristic of the parameter. This is in contrast with the self-reported belief about controllability, which was not correlated between the conditions (r=–0.18, p=0.26; *Figure 3f*; correlation between expected influence and self-reported controllability is listed in *Figure 4—figure supplement 4a-d*). Furthermore, we observed a positive association between expected influence and task performance during the Controllable condition (r=0.78, p<<0.0001; *Figure 3g*). This result suggests that those who simulated a greater level of controllability were able to raise the offers higher, indicating the beneficial effect of doing so.

## Comparison with a non-social controllability task

To investigate whether our results are specific to the social domain, we ran a non-social version of the task in which participants (n=27) played the same game with the instruction of 'playing with computer' instead of 'playing with virtual human partners.' Using the same computational models, we found that not only participants exhibited similar choice patterns (*Figure 2—figure supplement 1a-c*), but also the 2-step FT model was still favored in the non-social task (*Figure 2—figure supplement 1d,e*) and that delta was still higher for the Controllable than the Uncontrollable condition (*Figure 2—figure supplement 1f*, mean$_C$=1.31, mean$_U$=0.75, t(26)=2.54, p<0.05).

Interestingly, a closer examination of subjective data revealed two interesting differences in the non-social task compared to the social task. First, participants' subjective report of controllability did not differentiate between conditions in the non-social task (*Figure 2—figure supplement 1g*; mean$_C$=62.7, mean$_U$=56.9, t(25)=0.78, p=0.44), which suggests that the social aspect of an environment might have a unique effect on subjective beliefs about controllability. Second, inspired by

previous work demonstrating the impact of reward prediction errors (RPEs) on emotional feelings (*Rutledge et al., 2014*), we examined the impact of norm PE (nPE) on emotion ratings for the non-social and social contexts using a mixed effect regression model (*Supplementary file 1a*). We found a significant interaction between social context and nPE ($\beta$=0.52, p<0.05), suggesting that the non-social context reduced the impact of nPE on emotional feelings. Taken together, these new results suggest that despite of a similar involvement of FT in exploiting controllability, the social context had a considerable impact on subjective experience during the task.

## Replication of behavioral and computational findings in a large-scale online study

To test replicability and generalizability of our findings, we recruited 1342 participants from Prolific (http://prolific.co), an online survey platform, and had them play the game online (see Materials and methods for details). Notably, this online sample was more demographically diverse than the fMRI 'healthy' control, because we recruited them without any pre-screening or geographical constraints within the United States. Despite the greater level of diversity, the three model-agnostic findings remained robust. First, we found that the offer size increased throughout the trials under the Controllable condition, replicating the results from the fMRI sample (mean$_C$=6.0, mean$_U$ 5.0, t(1,341)=20.29, p<<0.0001; *Figure 4a*). Second, the rejection pattern was different between the two conditions, with a more flattened rejection curve for the Controllable than for the Uncontrollable condition (low ($1–3): mean$_C$=66%, mean$_U$=86%, t(741.54)=−12.28, p<<0.0001; middle ($4–6): mean$_C$=67%, mean$_U$=59%, t(2,606)=5.96, p<<0.0001; high ($7–9): mean$_C$=47%, mean$_U$=15%, t(1,925)=31.67, p<<0.0001; *Figure 4b*). Specifically, the online participants rejected more medium and high offers under the Controllable than the Uncontrollable, similar to the fMRI participants. Furthermore, for low offers, online participants showed significantly lower rejection rates under the Controllable than the Uncontrollable condition, which trend was not statistically significant for the fMRI sample. Third, online participants reported higher perceived control for the Controllable than the Uncontrollable as fMRI participants did (mean$_C$=58.3, mean$_U$=25.6, t(2,579)=27.93, p<<0.0001; *Figure 4c*).

Next, we tested whether the 2-step FT model performs as well for the large online sample as for the fMRI sample. First, we assessed the accuracy rate of the 2-step FT model's choice prediction; the mean of which was 80.2% for the Controllable and 93.9% for the Uncontrollable condition (*Figure 4—figure supplement 1a*). The parameters of the 2-step FT model were identifiable for the online sample as well (*Figure 4—figure supplement 1b-k*). Not only the model performance, but also the individual estimation results revealed consistency between the two heterogeneous samples. The parameter estimates for the online sample were comparable with the fMRI sample as shown in *Table 1*. The expected influence drawn from the 2-step FT model was higher for the Controllable than the Uncontrollable (mean$_C$=1.34, mean$_U$=0.90, t(1,341)=12.97, p<<0.0001; *Figure 4d*). Yet, consistent with the fMRI sample, the parameters for the two conditions were correlated (r=0.18, p<<0.0001; *Figure 4e*). The self-reported controllability showed a negative correlation between the conditions (r=−0.10, p<0.001; *Figure 4f*). In addition, the expected influence was positively correlated with the mean offer size (r=0.50, p<<0.0001; *Figure 4g*). Taken together, our independent large-scale replication results show that our suggested future thinking model explains decision processes involved in social controllability of a general population.

## The vmPFC computed summed choice values from the 2-step FT model

A computational model that could explain cognitive processes should not only fit choice behavior well, but also be represented by neurobiological substrates in the brain (i.e., biological plausibility) (*Cohen et al., 2017*; *O'Doherty et al., 2007*; *Wilson and Collins, 2019*). Accordingly, we expected that the total (both current and future) choice values estimated by the 2-step FT model, but not those from the 0-step model (only CVs), would be signaled in the vmPFC, a brain region that is known to process subjective values (*Bartra et al., 2013*; *Hiser and Koenigs, 2018*) during both social and non-social decision-making. To test this hypothesis, we regressed at the individual level trial-by-trial simulated normalized total values (TVs) of the chosen option drawn from the 2-step FT model (or the 0-step model in a separate GLM) as parametric modulators against event-related blood-oxygen-level-dependent (BOLD) responses recorded during fMRI (see Materials and methods). These analyses showed that the BOLD signals in the vmPFC tracked the value estimates drawn from the

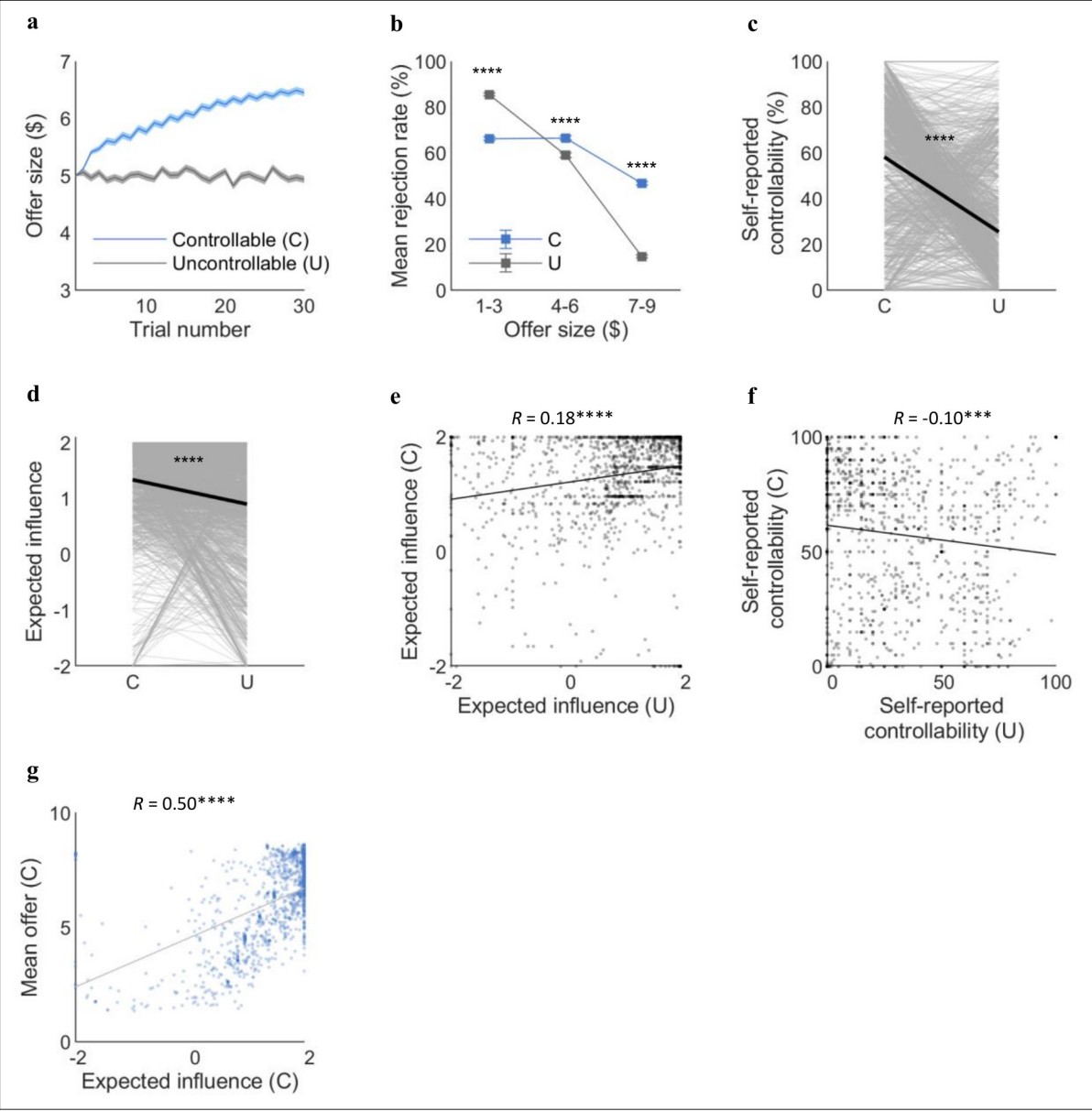

**Figure 4.** Replication of the behavioral and computational results in an independent large online sample (n=1342). (**a**) Online participants successfully increased the offer under the Controllable condition as fMRI participants did (mean_C=6.0, mean_U=5.0, t(1,341)=20.29, p<<0.0001). (**b**) Rejection rates binned by offer sizes differed between the two conditions in the online sample (low ($1–3): mean_C=66%, mean_U=86%, t(741.54)=–12.28, p<<0.0001; middle ($4–6): mean_C=67%, mean_U=59%, t(2,606)=5.96, p<<0.0001; high ($7–9): mean_C=47%, mean_U=15%, t(1,925)=31.67, p<<0.0001). (**c**) Online participants reported higher self-reported controllability for the Controllable than Uncontrollable (mean_C=58.3, mean_U=25.6, t(2,579)=27.93, p<<0.0001). (**d**) Consistent with the fMRI sample, expected influence was higher for the Controllable than the Uncontrollable for the online sample (mean_C=1.34, mean_U=0.90, t(1,341)=12.97, p<<0.0001). (**e**) The expected influence was correlated between the two conditions (r=0.18, p<<0.0001). (**f**) The self-reported controllability showed negative correlation between the two conditions for the online sample (r=–0.10, p<0.001). (**g**) The significant correlation between expected influence and mean offers under the Controllable was replicated in the online sample (r=0.50, p<<0.0001). Each dot represents a participant. The t-statistics for the mean offer size, binned rejection rate, and self-reported controllability are from two-sample t-tests assuming unequal variance using Satterthwaite's approximation according to the results of F-tests for equal variance. Error bars and shades represent SEM. For (**c**, **d**), each line represents a participant and each bold line represents the mean. C, controllable; U, Uncontrollable.

The online version of this article includes the following figure supplement(s) for figure 4:

**Figure supplement 1.** Model accuracy and parameter recovery for the online sample.

**Figure supplement 2.** Cross-parameter correlations.

**Figure supplement 3.** Online sample: results without those who had negative deltas.

**Figure supplement 4.** Correlations between expected influence and self-reported controllability for each condition and each sample.

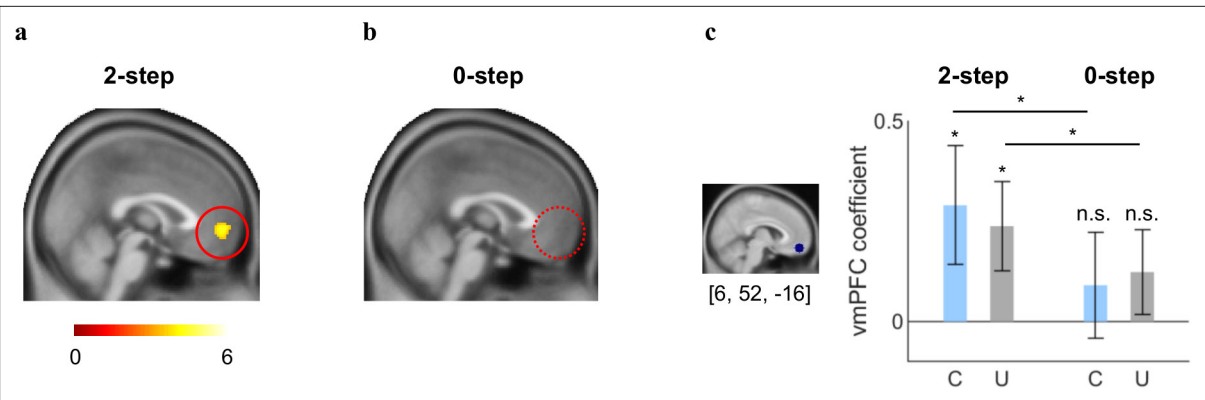

**Figure 5.** The ventromedial prefrontal cortex (vmPFC) computes projected summed choice values in exerting social controllability. (**a**) The vmPFC parametrically tracked mentally simulated values of the chosen actions drawn from the 2-step forward thinking (FT) model in both conditions ($P_{FDR}$ <0.05, k>50). (**b**) No activation was found in the brain including the vmPFC in relation with the value signals estimated from the 0-step model at a more liberal threshold (p<0.005, uncorrected, k>50). (**c**) The vmPFC ROI coefficients for the 2-step FT's value estimates were significantly greater than 0 for both the Controllable and Uncontrollable conditions (Controllable: $mean_C$=0.29, t(47)=1.96, p<0.05 (one-tailed); Uncontrollable: $mean_U$=0.24, t(47)=2.14, p<0.05 (one-tailed)) whereas the coefficients from the same ROI for 0-step's value estimates were not significant for either condition (Controllable: $mean_C$=0.09, t(46)=0.69, p=0.25 (one-tailed); Uncontrollable: $mean_U$=0.12, t(46)=1.17, p=0.12 (one-tailed)). The vmPFC coefficients were significantly higher under the 2-step model than the 0-step model for both the Controllable and Uncontrollable conditions (Controllable: t(46)=1.81, p<0.05 (one-tailed); Uncontrollable: t(46)=2.04, p<0.05 (one-tailed)). The coefficients were extracted from an 8-mm-radius sphere centered at [6, 52, −16] based on a meta-analysis study that assessed neural signatures in the ultimatum game (*Feng et al., 2015*). Error bars represent SEM; *p<0.05; n.s. indicates not significant. C, Controllable; ROI, region-of-interest; U, Uncontrollable.

The online version of this article includes the following figure supplement(s) for figure 5:

**Figure supplement 1.** Neural encoding of value in the vmPFC is associated with behavior-belief disconnect under the Uncontrollable condition.

**Figure supplement 2.** Current and future value signals.

**Figure supplement 3.** GLM comparison at the neural level.

**Figure supplement 4.** Norm prediction error signals.

**Figure supplement 5.** Norm signals.

2-step planning model across both conditions ($P_{FDR}$ <0.05, k>50; *Figure 5a*, *Supplementary file 1e*), and there was no significant difference between the two conditions ($P_{FDR}$ <0.05). In contrast, BOLD responses in the vmPFC did not track the trial-by-trial value estimates from the 0-step model, even at a more liberal threshold (p<0.005 uncorrected, k>50; *Figure 5b*, *Supplementary file 1f*). We also conducted model comparison at the neural level using the MACS toolbox (see *Figure 5—figure supplement 3* for details) and found that the vmPFC encoded TVs rather than only CV or FV.

These whole-brain analyses results were further corroborated by a set of independent region-of-interest (ROI) analyses. Specifically, we created a vmPFC ROI based on the peak coordinate from an independent meta-analysis on social decision making (an 8-mm-radius sphere centered at [6, 52, −16]) (*Feng et al., 2015*) and extracted parameter estimates from the mask. This analysis showed that the vmPFC ROI coefficients for the choice values were significantly greater for the 2-step model than for the 0-step model regardless of the condition (Controllable: t(46)=1.81, p<0.05 (one-tailed); Uncontrollable: t(46)=2.04, p<0.05 (one-tailed); *Figure 5c*). Indeed, the ROI coefficients based on the 2-step model were significantly larger than zero for each condition (Controllable: $mean_C$=0.29, t(47)=1.96, p<0.05 (one-tailed); Uncontrollable: $mean_U$=0.24, t(47)=2.14, p<0.05 (one-tailed); *Figure 5c*) whereas these coefficients for the choice values (CV only) based on the 0-step model were not significant for either condition (Controllable: $mean_C$=0.09, t(46)=0.69, p=0.25 (one-tailed); Uncontrollable: $mean_U$=0.12, t(46)=1.17, p=0.12 (one-tailed); *Figure 5c*). These findings suggest that individuals engaged the vmPFC to compute the projected total (current and future) values of their choices during FT. Furthermore, vmPFC signals were comparable between the two conditions both in the whole-brain analysis and the ROI analyses. Consistent with our behavioral modeling results, these neural results further support the notion that humans computed summed choice values regardless of the actual controllability of the social environment.

In addition, we examined whether norm prediction errors (nPEs) and norm estimates themselves from the 2-step FT model were tracked in the brain. We found that nPEs were encoded in the ventral striatum (VS; [4, 14, −14]) and the right anterior insula (rAI; [32, 16, −14]) for the Controllable condition (*Figure 5—figure supplement 4a*), while these signals were found in the anterior cingulate cortex (ACC; [2, 46, 16]) for the Uncontrollable condition (*Figure 5—figure supplement 4b*) at $P_{FWE}$ <0.05, small volume corrected. All three regions have been suggested to encode prediction errors in other norm learning tasks (*Xiang et al., 2013*). We further contrasted the whole-brain map of the two conditions and found that the VS ([4, 14, −14]) and the rAI ([32, 16, −14]) had significantly greater BOLD responses for the Controllable than the Uncontrollable condition ($P_{FWE}$ <0.05, small volume corrected; *Figure 5—figure supplement 4c*) whereas the ACC ([2, 46, 16]) response under the Uncontrollable condition was not significantly greater than the Controllable condition at the same threshold (*Figure 5—figure supplement 4d*). We also found that internal norm-related BOLD signals were tracked in the VS ([10, 16, −2]) for the Controllable condition (*Figure 5—figure supplement 4a*), and in the rAI ([28, 16, −6]) and the amygdala ([18, −6, −8]) for the Uncontrollable condition (*Figure 5—figure supplement 5b*) at $P_{FWE}$ <0.05, small volume corrected. However, the difference between the conditions was not statistically significant in the whole-brain contrast (*Figure 5—figure supplement 5c-d*). Taken together, these results suggest that the controllability level of the social interaction modulates neural encoding of internal norm representation and adaptation, expanding our previous knowledge about the computational mechanisms of norm learning (*Gu et al., 2015*; *Xiang et al., 2013*).

Finally, in an exploratory analysis, we examined the behavioral relevance of these neural signals in the vmPFC beyond the tracking of trial-by-trial values. Recall that despite the significant activations of the vmPFC in both conditions, individuals still exhibited different levels of self-reported controllability and the expected influence. Furthermore, there was condition-dependent discrepancy between the self-reported controllability and the expected influence (*Figure 4—figure supplement 4a-d*). Thus, we examined whether neural encoding of value in the vmPFC might relate to this discrepancy depending on the controllability of the environment. To do this, we assessed the correlation between extracted parameter estimates from the vmPFC and the disconnection between the belief and the expected influence (i.e., the 'biased belief' computed by subtracting the normalized expected influence from the normalized self-reported controllability). We found that the correlation between vmPFC-encoded value signals and the belief-behavior disconnection was indeed dependent on the condition (difference in slope: Z=2.40, p<0.05). Specifically, vmPFC signals were positively correlated with the disconnection between self-reported controllability and expected influence in the uncontrollable environment (r=0.35, p<0.05; *Figure 5—figure supplement 1a*), but not in the controllable environment (r=–0.14, p=0.38; *Figure 5—figure supplement 1b*). These results suggest that the meaning of vmPFC encoding of value signals could be context-dependent—and that heightened vmPFC signaling in uncontrollable situations is related to overly optimistic beliefs about controllability.

## Discussion

For social animals like humans, it is crucial to be able to exploit the controllability of our social interactions and to consider the long-term social effects of our current choices. This study provides a mechanistic account for how humans identify and use social controllability. In two independent samples of human participants, we demonstrate that (1) humans are capable of exploiting the controllability of their social interactions and exert influence on social others when given the opportunity, and that (2) they do so by engaging a mental model of FT and calculating the downstream effects of their current social choices. By using model-based fMRI analytic approach, we demonstrate that the vmPFC represents combined signals of CV and FV during forward social planning; and that this neural value representation was positively associated with belief-behavior disconnect in the Uncontrollable condition. These findings demonstrate that people use vmPFC-dependent FT to guide social choices, expanding the role of this neurocomputational mechanism beyond subjective valuation.

FT is an important high-level cognitive process that is frequently associated with abstract reasoning (*Hegarty, 2004*), planning (*Szpunar et al., 2014*), and model-based control (*Constantinescu et al., 2016*; *Daw et al., 2011*; *Gläscher et al., 2010*; *Schuck et al., 2016*; *Wang et al., 2018*). Also known as prospection, FT has been suggested to involve four intertwined modes: mental simulation, prediction, intention, and planning (*Szpunar et al., 2014*). All four modes are likely to have taken place in our study, as our FT model implies that a social decision-maker mentally simulates social value

functions into the future, predicts how her action would affect the following offers from partners, sets a goal of increasing future offers, and plans steps ahead to achieve the goal. Future studies will be needed to disentangle the neurocomputational mechanisms underlying each of these modes.

Critically relevant to the current study, previous research suggests that humans can learn and strategically exploit controllability during various forms of exchanges with others (*Bhatt et al., 2010*; *Camerer, 2011*; *Hampton et al., 2008*; *Hula et al., 2015*). The current study is in line with this literature and expands beyond existing findings. Here, we show that humans can also exploit controllability and exert their influence even when interacting with a series of other players (as opposed to a single other player as tested in previous studies). Furthermore, our 2-step FT model captures the explicit *magnitude* of controllability in individuals' mental models of an environment, which can be intuitively compared to subjective, psychological controllability. Finally, our 2-step FT model simultaneously incorporates aversion to norm violation and norm adaptation, two important parameters guiding social adaptation (*Fehr, 2004*; *Gu et al., 2015*; *Spitzer et al., 2007*; *Zhang and Gläscher, 2020*). These individual- and social-specific parameters will be crucial for examining social deficits in various clinical populations in future studies.

Our key parameter from the FT model is δ, the expected influence that individuals would be mentally simulating during decision processes. We found that individuals who showed higher δ performed better in terms of achieving higher offers from their partners under the Controllable condition, suggesting a direct association between FT and performance in strategic social interaction. Although δ was higher in the Controllable than the Uncontrollable condition, one surprising finding is that people's behavior was better explained by the 2-step FT model than the 0-step no planning model even for the Uncontrollable condition. In addition, we did not find any significant differences in vmPFC encoding of controllability between the conditions. These results suggest that participants still expected some level of influence (controllability) over their partners' offers even when environment was in fact uncontrollable. Furthermore, δ was positively correlated between the conditions, indicating the stability of the mentally simulated controllability across situations within an individual. We speculate that people still attempted to simulate future interactions in uncontrollable situations due to their preference and tendency to control (*Leotti and Delgado, 2014*; *Shenhav et al., 2016*).

Our modeling result was corroborated by neural findings of simulated total choice value encoding in the vmPFC regardless of the actual social controllability of conditions. There are currently at least two distinct views about the role of the vmPFC. The first view considers the vmPFC to encode a generic value signal (e.g., the common currency; *Levy and Glimcher, 2012*), including the value of social information (*Behrens et al., 2008*; *Chung et al., 2015*) and anticipatory utility (*Iigaya et al., 2020*). An alternative theory suggests that the vmPFC represents mental maps of state space (*Schuck et al., 2016*) and of conceptual knowledge (*Constantinescu et al., 2016*), in addition to other 'map'-encoding brain structures such as the hippocampus (*O'keefe and Nadel, 1978*; *Tavares et al., 2015*) and entorhinal cortex (*Stensola et al., 2012*). Of course, these views of the vmPFC might not necessarily be in contradiction. Instead, the exact function of the vmPFC could depend on the specific setup of the task environment. In the particular case of our task, as explicit values are inherently embedded in each state (i.e., each interaction), the vmPFC computed a summed value of not only the current state, but also future states. That is, both types of computations could be required to calculate the total downstream values of current social choices in our experimental setup. We also found that vmPFC signal was amplified by illusory beliefs only when the social environment was uncontrollable (but not when environment was controllable), suggesting that the behavioral relevance of value-encoding in the vmPFC is context-dependent. Taken together, our neural results illustrate a role of the vmPFC in constructing the TVs (both CV and FV) of current actions as humans engaged in forward planning during social exchange; and that these vmPFC-encoded values signals can be counterproductive and relate to exaggerated illusory beliefs about controllability when environment does not allow control.

Given our results, it is compelling to design tasks that focus on the way that subjects learn the model of an environment (in our terms, acquiring a value for the parameter δ) in early trials or build complex models of their partners' minds (as in a cognitive hierarchy; *Camerer et al., 2004*). Indeed, even though, in our task, the straightforward model based on norm-adjustment characterized participants' behavior well, there are more sophisticated alternatives that are used to characterize interpersonal interactions, such as the framework of interactive partially-observable Markov decision processes

(*Gmytrasiewicz and Doshi, 2005*; *Gu et al., 2015*; *Xiang et al., 2012*). These might provide additional insights into the sorts of probing that our subjects presumably attempted in early trials to gauge controllability (and the ways this differs in both the Controllable and the Uncontrollable conditions between subjects who do and do not suffer from substantial illusions of control). The framework would also allow us to examine whether our subjects thought that their partners built a model of them themselves (as in theory-of-mind or a cognitive hierarchy; *Camerer et al., 2004*), which would add extra richness to the interaction, and allow us to capture individual trajectories regarding social interactions in a finer detail—if, for instance, our subjects might have become irritated (*Hula et al., 2015*) at their partners' unwillingness to respond to their social signaling under the Uncontrollable condition.

The current study has the following limitations. First, due to the nature of the study design (i.e., reduction in uncertainty within the sequence of offers might be an inherent feature to controllability), the distributions of overall offers were not completely matched between conditions and may affect individuals' belief about their controllability. We did not find evidence that uncertainty or autocorrelation affected the expected influence or self-reported controllability and that reduction in uncertainty might be an inherent feature to controllability (*Supplementary file 1g*). Still, future experimental designs which dissociate change in uncertainty from change in controllability may better address potentially different effects of controllability and uncertainty on choice behavior and neural responses. Second, the lack of clear instruction in different controllability conditions in our study may have affected the extent to which individuals exploit controllability and develop illusion of control. Future studies implementing explicit instructions might be better suited to examine controllability-specific behaviors and neural substrates.

In summary, the current study provides a mechanistic account for how people exploit the controllability of their social environment. Our finding expands the roles of the vmPFC and model-based planning beyond spatial and cognitive processes. The implications of these findings could be far-reaching and multifaceted, as the proposed model not only showcases how FT can help optimize normative social behavior, as often required during strategic social interaction (e.g., negotiation, reputation building, and social networking), but may also help us understand how aberrant computation of social controllability may contribute to mental health symptoms and deterioration of group cooperation and trust in future studies.

## Materials and methods
### MRI participants
The study was approved by the Institutional Review Board of the University of Texas at Dallas and the University of the Texas Southwestern Medical Center (S.N., V.G.F, and X.G.'s previous institute where data were collected). The sample size was computed by G*Power 3.1.9.4. assuming a paired two-tailed t-test with the effect size of 0.5, alpha of 0.05, and the power of 0.95 was 54. 56 healthy adults (38 female, age=27.3±9.2 years, 3 left-handed) were recruited in the Dallas-Fort Worth metropolitan area. Participants provided written informed consent and completed this study. Five participants were excluded due to behavior data loss caused by computer collapse, one participant was excluded due to fMRI data loss, one participant was excluded due to excessive in-scanner head motion, and one participant was excluded due to poor quality of parameter recovery. The final sample had 48 healthy adults (33 female, age=27.6 ±9.1 years, 3 left-handed). Participants were paid a reward randomly drawn from the outcomes of this task, in addition to their baseline compensation calculated by time and travel distance.

### Online participants
The study was approved by the Institutional Review Board at the Icahn School of Medicine at Mount Sinai. Participants were recruited from Prolific (http://prolific.co), an online survey platform. A total of 1499 adults (734 female, age=35.1±13.1 years) provided online consent and completed this study. The online participant data were part of a larger study examining social cognition. We excluded 14 participants because of duplication of their data files and 143 additional participants because they had flat responses (accepted all or rejected all offers) for all the rounds within at least one condition. The final sample had 1342 adults (649 female, age=34.5 ± 12.8 years; report of demographics excluded another 21 participants who typed in an incorrect ID for the demographics survey and whose task

data were intact but could not be linked to demographic data). Participants were paid 10% of the reward drawn from a random trial of this task, in addition to $7.25 of the baseline compensation and the bonuses from the tasks other than the current social exchange game, which were not part of this study.

## Experimental paradigm: laboratory version

We designed an economic exchange task to probe social controllability based on an ultimatum game. This task consisted of two blocks, each representing an experimental condition ('Controllable' vs. 'Uncontrollable'). In both conditions, participants were offered a split of $20 by a partner and decided whether to accept or reject the proposed offer from the partner. If a participant accepted the proposal, the participant and the partner split the money as proposed. If a participant rejected the proposal, both the participant and the partner received nothing. At the beginning of each block, participants were instructed that they would play the games with members of Team A or Team B. This instruction allows participants to perceive players in each block as a group with a coherent norm, rather than random individuals. However, participants were not told how the players in each team would behave so that participants would need to learn the action-offer contingency. There were 40 trials in each block (for fMRI participants). In 60% of the trials, participants were also asked to rate their feelings after they made a choice.

In the Uncontrollable condition, participants played a typical ultimatum game: the offers were randomly drawn from a truncated Gaussian distribution (μ=$5, σ=$1.2, rounded to the nearest integer, max=$8, min=$2) on the fly using the MATLAB function 'normrnd' and 'round.' Thus, participants' behaviors had no influence on the future offers. Importantly, in the Controllable condition, participants could increase the next offer from their partner by rejecting the current offer, or decrease the next offer by accepting the present offer in a probabilistic fashion (⅓ chance of ±$2, ⅓ chance of ±$1, ⅓ chance of no change; the range of the offers for Controllable was between $1 and $9 [inclusive]—the range was not matched for the two conditions by mistake) (*Figure 1b*). We designed this manipulation based on the finding that reputation plays a crucial role in social exchanges (*Fehr, 2004*; *King-Casas et al., 2005*; *Knoch et al., 2009*); thus, in a typical ultimatum game, accepting any offers (although considered perfectly rational by classic economic theories; *Becker, 2013*) will develop a reputation of being 'cheap' and eventually lead to reduced offers, while the rejection response can serve as negotiation power and will force the partner to increase offers. At the end of each condition, participants also rated how much control they perceived using a sliding bar (from 0% to 100%).

## Experimental paradigm: online version

For the online study, we revised certain perceptual features of the task in order to better maintain participants' attention and ensure data quality, while maintaining the main structure of the task (Appendix 1). First, we reduced the number of trials to 30 from 40, considering both the minimal need for modeling purpose as well as the initial finding that behaviors typically stabilize after only 5 trials or so. We also introduced avatars in addition to partners' names to make the online interactions more engaging, and made minor revisions to the instructions to further emphasize that participants might or might not influence their partners' offers (but still without telling them how they might influence the offers or which team might be influenced). Finally, to remove unintended inter-individuals variability in offers for the Uncontrollable condition, we pre-determined the offer amounts under Uncontrollable (offers=[$1, 1, 2, 2, 2, 3, 3, 3, 4, 4, 4, 4, 5, 5, 5, 5, 5, 5, 6, 6, 6, 6, 7, 7, 7, 8, 8, 8, 9, 9], mean=$5.0, std=$2.3, min=$1, and max=$9) and randomized the order of them.

## Computational modeling

We hypothesized that people would estimate their social controllability by using the consequential future outcomes to compute action values. To test this hypothesis, we constructed a FT value function with different horizons: zero to four steps of forward planning whereby zero-step represents the no FT model.

First, we assumed that participants correctly understood the immediate rules of the task as follows:

$$a_i \in \{0, 1\} \tag{1}$$

$$r_i = \left\{ \begin{array}{ll} 0 \text{ if } a_i = 0 \\ s_i \text{ if } a_i = 1 \end{array} \right\} \tag{2}$$

$a_i$ represents the action that a participant takes at the th trial where 0 representing rejection and one representing acceptance. $r_i$ is the reward a participant receives at the th trial depending on $a_i$. Participants receive nothing if they reject whereas they receive the offered amount, $s_i$, if they accept.

Similar to our previous work on norm adaptation (**Gu et al., 2015**), we assumed that people are averse to norm violations, defined as the difference between the actual offer received and one's internal norm/expectation of the offers. Thus, the subjective utility of the expected immediate reward was constructed as follows.

Here, $U$, the utility, is a function of the reward and $f$ (internal norm) at the $i$th trial. The internal norm, which will be discussed in detail in the next paragraph, is an evolving reference value that determines the magnitude of subjective inequality. $\alpha$ ('sensitivity to norm violation,' $0 \leq \alpha \leq 1$) represents the degree to which an individual is averse to norm violation. We assumed that if one rejected the offer and received nothing, aversion would not be involved as the individual already understood the task rule that rejection would lead to a zero outcome. Given that, if there is only one isolated trial, participants will choose to accept or reject by comparing $U(s_i, f_i)$ (because $r_i = s_i$ if one accepts an offer) and $U(0, f_i) = 0$ (because $r_i = 0$ if one rejects).

For the internal norm updating, as our previous study (**Gu et al., 2015**) showed that Rescorla-Wagner (RW) (**Sutton and Barto, 2018**) models fit better than Bayesian update models, we used RW norm updates to capture how people learn the group norm throughout the trials as follows.

$$f_i = f_{i-1} + \varepsilon (s_i - f_{i-1}) \tag{3}$$

Here, $\varepsilon$ is the norm adaptation rate ($0 \leq \varepsilon \leq 1$), the individual learning parameter that determines the extent to which the norm prediction error ($s_i - f_{i-1}$) is reflected to the posterior norm. The initial norm was set as a free parameter ($\$0 \leq f_0 \leq \$20$).

Next, we formulated internal valuation as follows.

$$\Delta Q_i = v|_{a_i=1} - v|_{a_i=0} \tag{4}$$

$\Delta Q$, the difference between the value of accepting $\Delta Q$ and the value of rejecting $\Delta Q$, determines the probability of taking either action at the th trial. Importantly, we incorporated forward thinking procedure in calculation of. For an n-step forward thinking model, was calculated as follows.

$$v|_{a_i} = U(r_i, f_i) + \sum_{j=1}^{n} \gamma^j \times U(\hat{E}(r_{i+j}|a_i, \underline{a}_{i+1}, ... \underline{a}_{i+j}), f_i) \tag{5}$$

$$\hat{E}(s_{k+1}) = \left\{ \begin{array}{ll} s_k + \delta & \text{if } a_k \text{ or } \underline{a}_k = 0 \\ \max(s_k - \delta, 1) & \text{if } a_k \text{ or } \underline{a}_k = 1 \end{array} \right\} \tag{6}$$

$$\underline{a}_k = \left\{ \begin{array}{ll} 1 & \text{if } U(\hat{E}(s_k), f_k) > 0) \\ 0 & \text{otherwise} \end{array} \right\} \tag{7}$$

Given a hypothetical action $a_i$ in the current ($i^{\text{th}}$) trial, $v$ is the sum of the expected future reward utility assuming simulated future actions, $\underline{a}$. We used the term $E$ to represent an expected value in individuals' perception and estimation. We assumed that in individual's FT, her hypothetical action at the future trial ($\underline{a}_k$) increases or decreases the hypothetical next offer ($\hat{E}(s_{k+1})$) by $\delta$ ('expected influence,' $-\$2 \leq \delta \leq \$2$). Here, we assumed symmetric change ($\delta$) for either action so the change applies to both rejection and acceptance with the same magnitude but in the opposite direction. Given the structure of the task, we restricted $|\delta| \leq \$2$ in inference. Note that the main behavioral results (statistical testing results in **Figures 2–4**) remain true even if we excluded the subjects who showed negative deltas (**Figure 3—figure supplement 4**, **Figure 4—figure supplement 3**). We assumed that individuals knew that offers would not go below $1 because an offer of $0 would make their choice (accept or reject) undifferentiable. Although actual offers had an upper limit ($9), we did not set any upper

limit for individuals' hypothetical offers because there is no evidence for individuals to reason so especially until they repeatedly encounter offers of $9, even in which case individuals might or might not rule out the possibility of getting offered above $9. We assumed that simulated future actions ($\underline{a}_k$) are deterministic, contingent on the subjective utility of the immediately following rewards ($U\left(\hat{E}\left(r_k\right), f_k\right)$); this is a form of 1-level reasoning in a cognitive hierarchy (*Camerer et al., 2004*). The FVs computed through expected influence were discounted by $\gamma$, the temporal discounting factor. We fixed $\gamma$ at 0.8, the empirical mean across the participants from one initial round of estimation, in order to avoid collinearity with the parameter of our interest, $\delta$.

We modeled the probability of accepting the offer using the softmax function as follows:

$$P_i(a_i = 1) = \frac{1}{1+e^{-\beta Q_i}} \tag{8}$$

Here, $\beta$ ('inverse temperature,' $0 \leq \beta \leq 20$) indicates how strictly people base their choices on the estimated value difference between accepting and rejecting. The lower the inverse temperature is, the more exploratory the choices are.

We fit the model to individual choice data for the middle trials (30 trials for the fMRI sample and 20 trials for the online sample), excluding the first and the last five trials. The first five trials were excluded because one might be still learning the contingency between their action and the outcomes. The last five trials were also excluded because during those trials, the room to increase the offers becomes smaller and thus, participants had less incentive to reject offers as the interactions were close to the end (*Gneezy et al., 2003*).

## fMRI data acquisition and pre-processing

Anatomical and functional images were collected on a Philips 3T MRI scanner. High-resolution structural images were acquired using the MP-RAGE sequence (voxel size=1 mm×1 mm×1 mm). Functional scans were acquired during the participants completed the task in the scanner. The detailed settings were as follows: repetition time (TR)=2000 ms; echo time (TE)=25 ms; flip angle=90°; 38 slices; voxel size: 3.4 mm×3.4 mm×4.0 mm. The functional scans were preprocessed using standard statistical parametric mapping (SPM12, Wellcome Department of Imaging Neuroscience; https://www.fil.ion.ucl.ac.uk/spm/) algorithms, including slice timing correction, co-registration, normalization with resampled voxel size of 2 mm×2 mm×2 mm, and smoothing with an 8 mm Gaussian kernel. A temporal high-pass filter of 128 Hz was applied to the fMRI data and temporal autocorrelation was modeled using a first-order autoregressive function.

## fMRI general linear modeling

To find the BOLD responses that are correlated with the value estimates from the 2-step FT model and the 0-step model, we conducted two separate GLMs for each model. We specified each GLM with a parametric modulator of the chosen actions' values estimated from the corresponding model, normalized within a subject, at the individual level using SPM12. The event regressors were (1) offer onset, (2) choice submission, (3) outcome onset, and (4) emotion rating submission of the Controllable and Uncontrollable conditions. The parametric modulator was entered at the event of choice submission. In addition, six motion parameters of each condition were included as covariates. After individual model estimation, we generated the contrast images of whole-brain coefficient estimates with the contrast weight of 1 to value estimates of both the Controllable and Uncontrollable conditions. At the group-level, we conducted a one-sample t-test of the aforementioned individual whole-brain contrast images at $P_{FDR}$ <0.05 and k>50. We also conducted cross-validate Bayesian model selection (cvBMS) at the neural level using the MACS toolbox in SPM (*Soch and Allefeld, 2018*) in order to confirm that the vmPFC encoded TVs rather than only CVs or FVs. We considered four different GLMs: (i) the GLM with TV (our original GLM), (ii) the GLM with both CV and FV without orthogonalization (CV & FV), (iii) the GLM with only CV, and (iv) the GLM with only FV. All value estimates were extracted from the 2-step FT model. We computed the cross-validated log model evidence (cvLME) for each model at the individual level and computed exceedance probability (EP) of each model at the group level. For the ROI analysis, the vmPFC ROI (a 8-mm-radius sphere centered at [6, 52, −16]) was chosen from

an independent meta-analysis study (*Feng et al., 2015*) in which the coordinate was presented as showing greater activation for fair offers than unfair offers in the ultimatum game context. ROIs were extracted using the MarsBaR toolbox (*Brett et al., 2002*).

## Acknowledgements

XG is supported by the National Institute on Drug Abuse (Grant numbers: R01DA043695 and R21DA049243) and the National Institute of Mental Health (Grant numbers: R21MH120789, R01MH124115, and R01MH122611). This study was supported by a faculty startup grant to XG from the University of Texas, Dallas (where XG previously worked). DC is supported by UNIST internal funding (1.180073.01) and the National Research Foundation of Korea [Grant number: NRF-2018R1D1A1B07043582]. VGF is funded by the Mental Illness Research, Education, and Clinical Center (MIRECC VISN 2) at the James J. Peter Veterans Affairs Medical Center, Bronx, NY. PD is supported by the Max Planck Society and the Alexander von Humboldt Foundation. The authors thank Jae Shin for building the task website. The data in this study were used in a dissertation as partial fulfillment of the requirements for a PhD degree at the Graduate School of Biomedical Sciences at Mount Sinai.

## Additional information

### Funding

| Funder | Grant reference number | Author |
| --- | --- | --- |
| National Institute on Drug Abuse | R01DA043695 | Xiaosi Gu |
| National Institute on Drug Abuse | R21DA049243 | Xiaosi Gu |
| National Institute of Mental Health | R01MH124115 | Xiaosi Gu |
| National Institute of Mental Health | R01MH123069 | Xiaosi Gu |
| Max Planck Society | | Peter Dayan |
| Alexander von Humboldt Foundation | | Peter Dayan |
| National Institute of Mental Health | R21MH120789 | Xiaosi Gu |
| National Institute of Mental Health | R01MH122611 | Xiaosi Gu |
| Ulsan National Institute of Science and Technology | 1.180073.01 | Dongil Chung |
| National Research Foundation of Korea | NRF-2018R1D1A1B07043582 | Dongil Chung |
| Mental Illness Research, Education, and Clinical Center (MIRECC VISN 2), James J. Peter Veterans Affairs Medical Center | | Vincenzo G Fiore |

The funders had no role in study design, data collection and interpretation, or the decision to submit the work for publication.

### Author contributions

Soojung Na, Conceptualization, Investigation, Funding acquisition, Investigation, Methodology; Dongil Chung, Conceptualization, Funding acquisition, Writing – review and editing; Andreas Hula, Funding acquisition, Writing – review and editing; Ofer Perl, Conceptualization; Jennifer Jung, Matthew Heflin, Sylvia Blackmore, Investigation; Vincenzo G Fiore, Writing – review and editing; Peter

Dayan, Software, Funding acquisition, Writing – review and editing; Xiaosi Gu, Conceptualization, Funding acquisition, Methodology, Writing – review and editing, Supervision, Writing - original draft, Writing - review and editing

## Author ORCIDs
Soojung Na http://orcid.org/0000-0003-2565-5524
Dongil Chung http://orcid.org/0000-0003-1999-0326
Ofer Perl http://orcid.org/0000-0002-3560-4344
Matthew Heflin http://orcid.org/0000-0002-0379-3582
Vincenzo G Fiore http://orcid.org/0000-0002-4865-5482
Peter Dayan http://orcid.org/0000-0003-3476-1839
Xiaosi Gu http://orcid.org/0000-0002-9373-987X

## Ethics
All fMRI participants provided written informed consent and all online participants provided online consent. The fMRI study was approved by the Institutional Review Board of the University of Texas at Dallas (IRB 15-77) and the University of the Texas Southwestern Medical Center (STU 072015-031) (S.N., V.G.F, and X.G.'s previous institute where data were collected). Analyses of the original fMRI data collected at UT Dallas were covered by a Data Use Agreement between UT Dallas and the Icahn School of Medicine at Mount Sinai (ISMMS) (#19C7073) and IRB protocol approved by the ISMMS (HS#: 18-00728). The online study was approved by the Institutional Review Board at the Icahn School of Medicine at Mount Sinai (determined exempt; IRB-18-01301).

## Decision letter and Author response
Decision letter https://doi.org/10.7554/eLife.64983.sa1
Author response https://doi.org/10.7554/eLife.64983.sa2

---

# Additional files

### Supplementary files
• Supplementary file 1. Supplementary tables.
• Supplementary file 2. Task instructions.
• Transparent reporting form

### Data availability
The fMRI and behavioral data and analysis scripts are accessible at https://github.com/SoojungNa/social_controllability_fMRI, (copy archived at swh:1:rev:8ea1fb4fe6cbd625f9a25fe292f82fc953f8c713).

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

# Appendix 1

## Task design for online study

The task was proceeded as shown in *Appendix 1—figure 1*.

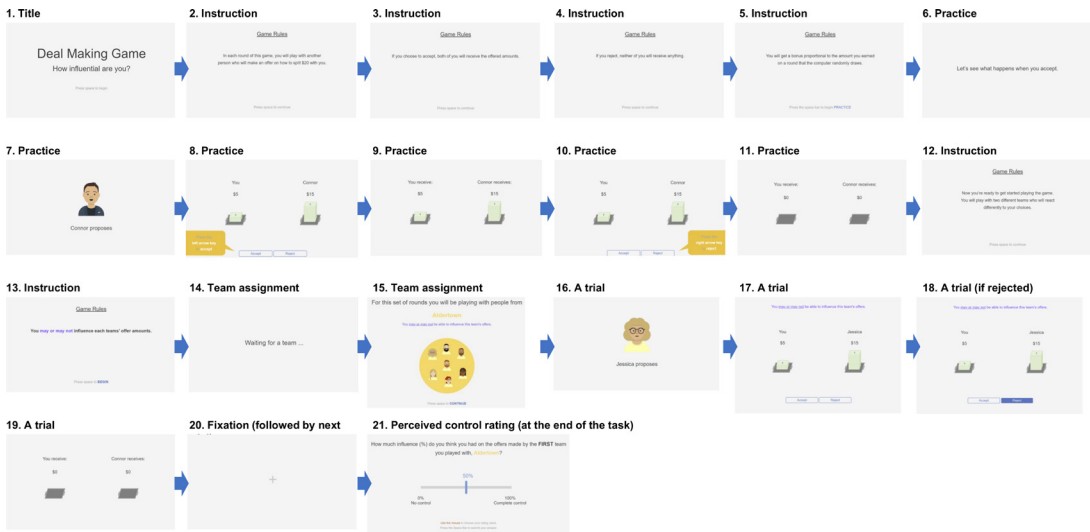

**Appendix 1—figure 1.** Task design for online study' and caption: Screen #6–11: Practice rounds. Screen #14–15: Team assignment. Displayed at the beginning of each condition. Screen #16–20: One round of the actual task; repeated 30 times for each team (condition). The order of partners (avatars and names) were randomized. Duration: Screen #16 (avatar): 1.5–2.5 s; jittered, screen #17 (choice): self-paced, screen #18 (post-choice), #19 (outcome), #20 (fixation): 1 s.

- Screen #6–11: Practice rounds.
- Screen #14–15: Team assignment. Displayed at the beginning of each condition.
- Screen #16–20: One round of the actual task; repeated 30 times for each team (condition).
  - The order of partners (avatars and names) were randomized.
  - Duration:
    - Screen #16 (avatar): 1.5–2.5 s; jittered
    - Screen #17 (choice): self-paced
    - Screen #18 (post-choice), #19 (outcome), #20 (fixation): 1 s

