## [Decision Letter]

**Acceptance summary:**

We believe that this examination of the neural and cognitive processes through which social controllability influences decision-making will be of interest to cognitive neuroscientists interested in the computational mechanisms involved in planning and social decision-making. The additional analyses and thorough revisions made to the manuscript have substantially strengthened the paper, and the conclusions and interpretations presented here are well supported by the data.

**Decision letter after peer review:**

Thank you for submitting your article "Humans Use Forward Thinking to Exploit Social Controllability" for consideration by *eLife*. Your article has been reviewed by 3 peer reviewers, including Catherine Hartley as the Reviewing Editor and Reviewer #1, and Christian Büchel as the Senior Editor. The following individual involved in review of your submission has agreed to reveal their identity: Romain Ligneul (Reviewer #3).

The reviewers have discussed the reviews with one another and the Reviewing Editor has drafted this decision to help you prepare a revised submission.

We would like to draw your attention to changes in our policy on revisions we have made in response to COVID-19 (https://elifesciences.org/articles/57162). Specifically, when editors judge that a submitted work as a whole belongs in eLife but that some conclusions require a modest amount of additional new data, as they do with your paper, we are asking that the manuscript be revised to either limit claims to those supported by data in hand, or to explicitly state that the relevant conclusions require additional supporting data.

Summary:

In this manuscript, Na and colleagues the examine the influence of perceived controllability on choice patterns in an ultimatum game task. In a task variant introducing both controllable and uncontrollable conditions, they find that endowing participants with controllability increases rejection rates in a way which enables social transaction to converge towards fairer offers. In order to clarify the cognitive underpinnings of this finding, the authors fit computational models that varied the degree to which the anticipation of future offers – controllable or not – could influence the decisions relative to current offers. Under the best fitting model, social controllability (the influence of current decision on future offers) is used to adjust the expected value of their decision to accept or reject offers, suggesting that subjects used a controllability-dependent forward planning process. The fMRI results suggest that the cognitive operations underlying this forward planning process might depend in part on computations within the ventromedial prefrontal cortex, in which BOLD activation correlated with total (current and future) values computed under such a model.

In most studies using the UG game studies, researchers look at how accept/reject behavior changes conditioned on the offer size, and the UG games are typically one-shot. Na et al., extend this work by looking at how the behavior of receivers could, in turn, affect future offers by the proposer, in an interactive manner. A further strength is that the authors were able to replicate the behavioral and modeling findings in a separate large online sample, and all data and analyses are made available online so that others could make use of them. Overall, the analyses are carefully performed and largely in support of the key conclusions. However, the reviewers felt that some aspects of the analysis could be further developed, refined, or clarified.

Revisions:

1) The background and rationale for the current study could be laid out more clearly in the introduction. The authors should explain what controllability means and why it is important. The introduction would also benefit from inclusion of some important behavioral and neural findings in the literature regarding controllability in non-social contexts. The discussion of "model-based planning" may not be so relevant here. In the 2-step task (Daw et al., 2011), participants need to learn a task transition structure and use this learned knowledge to plan future actions. But in the current task, there is no such abstract structure to learn. A discussion of the role of simulating future events/outcomes (e.g., counterfactual simulation) may be more appropriate than a focus on model-based planning. The authors may also want to include key studies and findings on strategic decision-making and theory of mind. The neural hypotheses should also be introduced, or if the authors didn't have a priori hypotheses, it could be explicitly stated that it is an exploratory study if indeed the case. If the vmPFC is indeed the area of interest a priori, then the authors should provide justification for this hypothesis.

2) It would be helpful to clearly assess and discuss the commonalities and differences in results between the social and non-social versions of the task, and their implications for interpreting the findings. It would be beneficial to see the computational model comparison applied to the non-social control experiment, as well (. Critically, is the 2-step model still favored.

3) The analysis of overall rejection rates (Figure 2b1) is slightly puzzling with respect to the results reported Figure 2a1 and 2b2. Indeed, Figure 2a1 shows that participants encountered a much higher proportion of middle and high offers in the controllable condition (due to their control over offers) and Figure 2b2 shows a very significant increase in rejection rates for these two types of offers but only a modest decrease for low offers. In addition, the offers in the uncontrollable condition seem to vary in a systematic fashion across time and to be very rarely below 3$. In this context, I wonder how mean rejection rates can possibly be equal across controllability conditions. Still regarding rejection rates, it also seems that the uncontrollable condition was associated with a much greater inter-individual variability in rejection rates, hence suggesting that controllability reduced variability in the type of strategy used to solve the task. The authors should (i) clarify how offers of the uncontrollable conditions were generated, (ii) discuss and perhaps try to explain (and relate to other findings) the different inter-individual variability in rejection rates across conditions.

4) In the behavioral analyses, what is the rationale for grouping the offer sizes into three bins rather than using the exact levels of offer sizes? Do the key results hold if exact values are used?

5) It would be helpful to include an analysis of response times. Indeed, one would expect forward planning to be associated with lengthened decision times and correspondingly, for the δ parameter (or strategizing depth, or controllable condition) to be associated with longer decision times (e.g. Keramati et al., Plos Comp. Biol., 2011). Furthermore, it was recently shown that perceived task controllability increases decision times, even in the absence of forward value computations (Ligneul et al., Biorxiv). It is also good practice to include decision times as a control parametric regressor when analyzing brain activities related to a variable potentially correlated with them. Furthermore, one could expect longer reaction times for more conflicting decisions (i.e. closer valuations of reject/accept offers).

6) The authors refer to the δ parameter "modeled controllability", however the model doesn't provide any account of the process of estimating controllability from observed outcomes (see Gershman and Dorfman 2019, Nature Communications or Ligneul et al., 2020, Biorxiv for examples of such models), but only reflects the impact of controllability on value computations, or the monetary amount of "expected influence" in each condition. An augmented model might include a computation of controllability, with the δ parameter controlling the extent to which estimated controllability promotes forward planning. Even if the authors don't fit such a model, they should explicitly acknowledge that their algorithm does not implement any form of controllability estimation, and might consider calling δ a "forward planning parameter". In addition, it is unclear why the authors chose to constrain the δ parameter to fluctuate between -2 and 2$ (rather than between 0 and 2$, in line with their experimental design, or with even broader bounds) and what a negative δ would imply. Also, would it make sense to exclude participants with a negative δ in addition to those with a δ greater than 2? Do all results hold under these exclusions?

7) While the authors performed a parameter recovery analysis, they did not report cross-parameter correlations, which are important for interpreting the best-fitting parameters in each condition. Furthermore, it is good practice to perform model recovery analyses on top of parameter recovery analyses (Wilson and Collins, 2019, *eLife*; Palminteri et al., 2017, TiCS) in order to make sure that the task can actually distinguish the models included in the model comparison. As a result, the conclusions based on model comparison and parameters values (that is, a significant part of the empirical results) are uncertain. The cross-correlation between parameters and model recovery analysis should be reported as a confusion matrix.

8) The parameters of the adaptive social norm model exhibit fairly poor recoverability, particularly in the controllable condition. The motivation for using this model is that it provided the best fit to subjects data in a prior uncontrollable ultimatum game task, but perhaps such adaptive judgment is not capturing choice behavior well here. It would be helpful to see a comparison of this model with one that has a static parameter capturing each individual's subjective inequity norm.

9) The authors stated that future actions are deterministic (line 576) contingent on the utility following the immediate reward. If so, is Figure 3a still valid? If all future actions are deterministic, there should be only one path from the current to the future, rather than a tree-like trajectory.

10) The MF model, and the rationale for its inclusion in the set of models compared, needs to be explained more clearly. The MF model appears to include no intercept to define a base probability of accepting versus rejecting offers, which makes it hard to compare with the other models in which the initial norm parameter may mimic such an intercept.

11) The fact that the vmPFC encoded total future + current value (2-step) and not current value (0-step) suggests that it might be specifically involved in computing future values but the authors do not report directly the relationship between its activity and future values. How correlated are the values from the 0-step model and the 2-step model? And more importantly, if vmPFC is associated with TOTAL value but not the current value, should that mean the vmPFC is associated with the future value only? It might make more sense to decompose the current value and future value both from the winning 2-step model, and construct them into the same GLM without orthogonalization.

12) The vmPFC result contrast averages across the controllable and uncontrollable conditions (line 629). Why did the authors do so? Wouldn't it be better to see whether the "total value" is represented differently between the two conditions.

13) The analysis of the relation between the vmPFC β weights and the difference between self-reported controllability beliefs and model-derived controllability estimates (Figure 5 d and e) is not adequately previewed. The hypothesis for why vmPFC activity might track this metric is unclear. Moreover, the relation between the two in the uncontrollable condition is somewhat weak. The authors should report the relation between vmPFC β weights and each component of the difference score (modeled and self-report controllability), and clearly motivating their intuition for why vmPFC activation might be related to that metric. If the authors feel strongly that this analysis is important to include, it would be meaningful to see whether the brain data could help explain behavioral data. For example, a simple GLM could serve this purpose: mean_offer ~ β(vmPFC) + self-report_controllability + model_controllabilty. Note that the authors need to state the exploratory nature if they decide to run this type of analysis.

14) The authors might also report the neural correlates of the internal norm and the norm prediction error (line 544). If the participants indeed acquired the social controllability through learning, they might form different internal norms in the two conditions, hence the norm prediction error might also differ.

15) Specific aspects of the experimental design may have influenced the observed results in ways that were not controlled. For example, it is not only the magnitude and controllability of outcomes that differed between the controllable and uncontrollable conditions, but also the uncertainty. It is possible that the less variable offers encountered in the controllable condition may have driven some of the results. The authors should acknowledge that the possible role of autocorrelation and uncertainty on behavioral and modeling results.

16) Moreover, asking participants to repeatedly rate their perception of controllability almost certainly influenced and exacerbated the impact of this factor on choices. It would have been very useful to perform a complementary online study excluding these ratings to ensure that controllability-dependent effects are still evident in such a case.

[Editors' note: further revisions were suggested prior to acceptance, as described below.]

Thank you for resubmitting your work entitled "Humans Use Forward Thinking to Exploit Social Controllability" for further consideration by *eLife*. Your revised article has been evaluated by Christian Büchel (Senior Editor), Catherine Hartley (Reviewing Editor), and the two original reviewers.

As you will read, the reviewers are in agreement that your manuscript has been substantially strengthened by these revisions, but there are some remaining issues that need to be addressed.

A primary concern is that the manuscript does not provide sufficiently strong support for the claim that the vmPFC supports forward planning, particularly in light of the new neuroimaging analyses performed as part of this revision. Reviewer 3 has a concrete suggestion for how this claim might be strengthened with a model comparison analysis. If further evidence for the claim is not found/provided, it should be tempered. Reviewer 2 also questions whether it is useful and sensible to retain the MF model in the set of compared models, and both reviewers note a few areas where clarification, greater methodological detail, or further interpretation are warranted.

Please carefully consider each of the reviewers suggestions as you revise your manuscript.

*Reviewer #2:*

The authors have revised their manuscript considerably and addressed a number of concerns raised in the initial review, with their additional analyses and detailed clarification. I particularly appreciate that the authors took the courage to dive into the direct comparison of findings between the social and non-social groups, which provided new insights. Furthermore, the revised Introduction is more thought-provoking with relevant literature included. Now the conclusions are better supported as it stands, and these findings are certainly going to be exciting additions to the literature of social decision neuroscience.

Here I have a few additional points, more for clarification.

(1) In response to comment #2, the authors might unpack the significant interaction result, to explicitly show "that the non-social context reduced the impact of nPE on emotional feelings." Also in the same LME model, I am curious about the significant "Controllable × social task (***)" interaction (β = -5.06). Does this mean, being in the Controllable + Social group, the emotion rating is lower? How would the authors interpret this finding?

(2) In response to comment #5 regarding response time with the additional LME analyses, I wonder which distribution function was used? We know that RT data is commonly positively skewed, so a log-normal or a shifted log-normal should be more accurate.

(3) I retain my initial comment regarding the inclusion of the MF model. The task is deterministic – participants get what appears if they accept and 0 if reject. In fact, the model is making a completely different prediction: according to the Q-value update, if the participant chose an "accept" and then indeed received a reward, then they should repeat "accept". But in the current task design, such a "positive feedback" would make the participants feel they are perhaps too easy to play with, and will be more likely to choose "reject" on the next trial. In essence, the MF model is not even capturing the behavioral pattern of the task, hence it does not seem to be a good baseline model. Rather, the 0-step model is okay enough to be the reference model.

*Reviewer #3:*

The authors have made very significant efforts to respond to a diversity of concerns and to amend their paper accordingly. The revised version is thus more complete and I believe that the main argument of the paper has been made stronger.

In many cases, the authors have appropriately adjusted their language in order to better align their conclusions with the data (e.g. renaming the δ parameter expected influence parameter) and I think that this paper can constitute an interesting addition to the field.

However, I am still slightly skeptical about the reach of neuroimaging results and I believe that some limitations of the paradigm may be more explicitly discussed.

A. Neuroimaging.

The authors have performed valuable additional analyses regarding the norm and norm prediction errors signals which can be of interest for the field. But I believe that our main concerns about vmPFC effects have not been fully addressed. Indeed, the authors still write that the vmPFC constructs "the total values (both current and future) of current actions as humans engaged in forward planning during social exchange". However, when splitting the analysis of current and future values, the encoding of future values was found in the insula whereas the vmPFC only encoded current values. The authors claim that the lack of encoding of total values derived from the 0-step FT model constitutes evidence in favor of forward planning, but it could be that this lack of evidence is driven by a poorer fit of current (rather than total) values by this simpler model. In order to better substantiate their claim about vmPFC's role, the authors may want to perform a model comparison at the neural level by comparing GLMs (using for example the MACS toolbox) including current value only, current value and future value, future value only or total value. Alternatively, they could analyze the first-level residuals produced by GLMs including alternatively current value, future value and total value (all based on FT-2). If their interpretation is correct, GLMs equipped with a parametric regressor for total value should be associated with smaller residuals in the vmPFC.

Regarding the behavior-belief disconnection analysis, I think that it would be more sensical to study the ratio rather than the difference between behavior and subjective reports, since these two measures are qualitatively different. Finally, it might be worth providing the reader with a brief discussion of the other neural substrates uncovered by the most recent analyses (dmPFC, insula, striatum, etc.).

B. Behavioral paradigm.

I believe that the authors should provide a few more details in the methods and acknowledge a few limitations in their discussion.

First, unless I am mistaking the method used to decide on block order (i.e. C or U first) was not reported. Was the "illusion of control" in the uncontrollable condition driven by the subset of participants who passed the controllable block first? If this is the case, then it might add some plausibility to the interpretation of subjective controllability ratings in the uncontrollable condition as an "illusion of control" (persistence of a control prior). In other words, I think that the authors should refrain from interpreting the raw value of these ratings as an illusion of control (perhaps not all participants understood the meaning of the rating, perhaps they were too lazy to move the cursor until 0, etc.).

While it does not necessarily implies an illusion of control, the fact that participants still relied on on forward planning in the uncontrollable condition (as indexed by the expected value parameter) is presumably what prevented authors to really isolate the neural substrates of strategic controllability-dependent forward planning, and it might thus be mentioned as a limitation of the paradigm.

I believe that it is also important to mention explicitly the fact that a third and a quarter of the data was excluded from the analyses of behavioral and fMRI data (i.e. first and last five trials of each block) respectively and the rationale for this exclusion may be discussed.

The authors wrote that "a task that carefully controls for uncertainty and autocorrelation confounds would help better understanding the accumulative effects of social controllability", which is a good start, but it would be in my opinion important to explicitly acknowledge that change in controllability were confounded with change in uncertainty about upcoming offers.

I would be curious to hear the authors' insight about why participants in the online study (and to some extent in the lab) accepted more often the low offers in the controllable condition. It seems somehow counterintuitive and could mean that participant behaved in a more "automatic" and perseverative way in the controllable condition.

Related to this last point, is it possible that the δ parameter (or expected influence) simply captures a perseverative tendency in rejection/acceptance of offers? This might explain the disconnection between behavior and belief, as well as the positive value of this parameter in the uncontrollable condition, correlated to that of the controllable one. That perseveration increases in the controllable condition would be logical (since that condition allows participants to reach their goal by doing so) and it would therefore still be of interest in the context of this social controllability study. Perhaps the authors could exclude this possibility by running adding a perseveration mechanism to their model, as it is often done in the RL literature?

---

## [Author Response]

Revisions:1) The background and rationale for the current study could be laid out more clearly in the introduction. The authors should explain what controllability means and why it is important. The introduction would also benefit from inclusion of some important behavioral and neural findings in the literature regarding controllability in non-social contexts. The discussion of "model-based planning" may not be so relevant here. In the 2-step task (Daw et al., 2011), participants need to learn a task transition structure and use this learned knowledge to plan future actions. But in the current task, there is no such abstract structure to learn. A discussion of the role of simulating future events/outcomes (e.g., counterfactual simulation) may be more appropriate than a focus on model-based planning. The authors may also want to include key studies and findings on strategic decision-making and theory of mind. The neural hypotheses should also be introduced, or if the authors didn't have a priori hypotheses, it could be explicitly stated that it is an exploratory study if indeed the case. If the vmPFC is indeed the area of interest a priori, then the authors should provide justification for this hypothesis.

Thank you for this suggestion. We fully agree that the background and rationale for the study could be more clearly laid out. We have now re-written both Introduction and Discussion sections to include literature more relevant to (non-social) controllability (e.g. Huys and Dayan, 2009), as well as future simulation (e.g. Szpunar et al., 2014), strategic decision-making (e.g. Hampton et al., 2008, Bhatt et al., 2010), and theory of mind (e.g. Hula et al., 2015) instead of focusing exclusively on model-based planning (whilst noting that simulating potential future outcomes is a prominent method of model-based planning in both artificial and natural systems). Furthermore, we have re-constructed the paragraphs about our neural hypothesis and why the vmPFC was our region of *a priori* interest.

Line 16: “Based on previous work demonstrating the computational mechanisms of controllability in non-social environments, here we hypothesize that people use mental models to track and exploit social controllability, for instance via forward simulation. In non-social contexts, it has been proposed that controllability quantifies the extent to which the acquisition of outcomes, and particularly desired outcomes, can be influenced by the choice of actions (Huys and Dayan, 2009; Dorfman and Gershman, 2019; Ligneul, 2021). […] Lastly, we hypothesize that the choice values integrating the planned paths would be signaled in the vmPFC.”

Line 399: “Critically relevant to the current study, previous research suggests that humans can learn and strategically exploit controllability during various forms of exchanges with others (Bhatt et al., 2010; Camerer, 2011; Hampton et al., 2008; Hula et al., 2015). The current study is in line with this literature and expands beyond existing findings. Here, we show that humans can also exploit controllability and exert their influence even when interacting with a series of other players (as opposed to a single other player as tested in previous studies). Furthermore, our 2-step FT model captures the explicit *magnitude* of controllability in individuals’ mental models of an environment, which can be intuitively compared to subjective, psychological controllability. Finally, our 2-step FT model simultaneously incorporates aversion to norm violation and norm adaptation, two important parameters guiding social adaptation (Fehr, 2004; Gu et al., 2015; Spitzer et al., 2007; Zhang and Gläscher, 2020). These individual- and social- specific parameters will be crucial for examining social deficits in various clinical populations in future studies.”

2) It would be helpful to clearly assess and discuss the commonalities and differences in results between the social and non-social versions of the task, and their implications for interpreting the findings. It would be beneficial to see the computational model comparison applied to the non-social control experiment, as well (. Critically, is the 2-step model still favored.

Thank you for this comment. Following your suggestion, we have now applied our computational models to the non-social version of the task in a new set of analyses. These new analyses revealed overlapping, yet distinct mechanisms of social and non-social controllability, as detailed below.

First, we found that the 2-step FT model was still favored in the non-social task. We also found that the estimated δ was still higher for the Controllable condition than the Uncontrollable condition in the non-social version of the task (see new Figure 2—figure supplement 1). These findings suggest that forward thinking could be a fundamental mechanism for humans to exert control across social and non-social domains.

Second, we discovered several striking differences in people’s subjective states between social and non-social contexts. As already reported in the original paper, subjective beliefs about controllability were significantly different between the social and non-social contexts. When playing against computer algorithms (i.e., non-social context), participants reported a similar level of perceived controllability (~50%) for both the Controllable and Uncontrollable conditions (Figure 2c), albeit they, evidenced by model-based analyses, mentally simulated a higher level of control in the Controllable condition than in the Uncontrollable condition. This is in sharp contrast to the social task results where participants reported perceiving higher controllability for the Controllable condition (65.9%) compared to the Uncontrollable condition (43.7%) (difference of the self-reported controllability between the conditions (Controllable – Uncontrollable) : mean_social task_ = 22.1; mean_non-social task_ = 0.3; paired t-test *t*(19) = -2.98, *P* <.01; Figure 2-figure supplement 1g).

Third, another major difference between the social and non-social contexts was observed in people’s emotional ratings. Previously, it was demonstrated that reward prediction errors (PE) experienced by individuals were significantly associated with the trajectories of self-reported emotional feelings (Rutledge et al., 2014). Building on this finding and our previous finding of how the non-social context reduced subject reports of control, we hypothesized that this emotional engagement would be modulated by the social context, such that the association between PE and emotional ratings would be weaker in the non-social than in the social context. To address this, in our new analysis, we ran a mixed effect GLM predicting emotion ratings with norm prediction errors, task types (social and non-social), interactions between the norm prediction errors and the task types, and other controlling variables such as offers, conditions, and individual random effects ('emotion rating ~ offer + norm prediction error + condition + task + task*(offer + norm prediction error + condition) + (1 + offer + norm prediction error | subject)'). As expected, the impact of norm prediction errors on happiness ratings was reduced in the non-social context compared to the social context (Supplementary file 1a; significant interaction of nPE x social context).

Taken together, these new results demonstrate both overlapping and distinct processes involved in social vs. non-social controllability. Despite a similar involvement of forward thinking in choice behaviors, we speculate that participants might have considered a computer algorithm to be more objective than a human player. Therefore, despite their similar ability to make choices and influence future outcomes, participants did not perceive the Controllable condition to be more controllable than the Uncontrollable condition; nor did they consider the PE signals to be as impactful on their feelings of happiness in the non-social condition as in the social condition. This set of findings further support the notion that subjective states could be detached from action or planning per se; and that the social context modulates the relationship between subjective states and choices.

We have now added all these results and discussion points to the revised manuscript.

Line 253: “Comparison with a non-social controllability task. To investigate whether our results are specific to the social domain, we ran a non-social version of the task in which participants (n=27) played the same game with the instruction of “playing with computer” instead of “playing with virtual human partners”. Using the same computational models, we found that not only participants exhibited similar choice patterns (Figure 2—figure supplement 1a-c), but also the 2-step FT model was still favored in the non-social task (Figure 2—figure supplement 1d,e) and that δ was still higher for the Controllable than the Uncontrollable condition (Figure 2—figure supplement 1f**,** mean_C_ = 1.31, mean_U_ = 0.75, *t*(26) = 2.54, *P* < 0.05).

Interestingly, a closer examination of subjective data revealed two interesting differences in the non-social task compared to the social task. First, participants’ subjective report of controllability did not differentiate between conditions in the non-social task (Figure 2—figure supplement 1g; mean_C_ = 62.7, mean_U_ = 56.9, *t*(25) = 0.78, *P* = 0.44), which suggests that the social aspect of an environment might have a unique effect on subjective beliefs about controllability. Second, inspired by previous work demonstrating the impact of reward prediction errors (PE) on emotional feelings (Rutledge et al., 2014), we examined the impact of norm PE (nPE) on emotion ratings for the non-social and social contexts using a mixed effect regression model (Supplementary file 1a). We found a significant interaction between social context and nPE (β = 0.52, *P* < 0.05), suggesting that the non-social context reduced the impact of nPE on emotional feelings. Taken together, these new results suggest that despite of a similar involvement of forward thinking in exploiting controllability, the social context had a considerable impact on subjective experience during the task.”

3) The analysis of overall rejection rates (Figure 2b1) is slightly puzzling with respect to the results reported Figure 2a1 and 2b2. Indeed, Figure 2a1 shows that participants encountered a much higher proportion of middle and high offers in the controllable condition (due to their control over offers) and Figure 2b2 shows a very significant increase in rejection rates for these two types of offers but only a modest decrease for low offers. In addition, the offers in the uncontrollable condition seem to vary in a systematic fashion across time and to be very rarely below 3$. In this context, I wonder how mean rejection rates can possibly be equal across controllability conditions. Still regarding rejection rates, it also seems that the uncontrollable condition was associated with a much greater inter-individual variability in rejection rates, hence suggesting that controllability reduced variability in the type of strategy used to solve the task. The authors should (i) clarify how offers of the uncontrollable conditions were generated, (ii) discuss and perhaps try to explain (and relate to other findings) the different inter-individual variability in rejection rates across conditions.

Thank you for these comments. Per your suggestion, we now clarify how the offers were generated for the Uncontrollable condition and discuss the different variance in rejection rates across conditions.

First, the offers were randomly drawn from a truncated Gaussian distribution (μ = $5, σ = $1.2, min = $2, max = $8; on the fly rather than predetermined) in the Uncontrollable condition for the fMRI sample. As a result, individuals had a slightly different set of offers for the Uncontrollable condition as well as for the Controllable condition where people’s controllability level differed. That is, the number of trials in each offer bin differed by bins and individuals. To calculate the binned rejection rates of our entire sample (Figure 2b2), we first calculated the mean rejection rate for each bin, per each individual, and then aggregated across all individuals. This approach rather than aggregating all the trials across all individuals at once gives the same weight to each individual within a binned subsample. Thus, the depicted overall rejection rates (the thick line in Figure 2b1) is different from the simple average of the binned rejection rates. All these data are made available through a repository (https://github.com/SoojungNa/social_controllability_fMRI) for any readers who might be interested in further exploring. We have now added these clarifications in the revised manuscript.

Line 132: “Next, we examined the rejection patterns from the two conditions. On average, rejection rates in the two conditions were comparable (mean_C_ = 50.8%, mean_U_ = 49.1%, *t*(67.87) = 0.43, *P* = 0.67; Figure 2b1). By separating the trials each individual experienced into three levels of offer sizes (low: $1-3, medium: $4-6, and high: $7-9) and then aggregating across all individuals, we further examined whether rejection rates varied as a function of offer size. We found that participants were more likely to reject medium to high ($4-9) offers in the Controllable condition, while they showed comparable rejection rates for the low offers ($1-3) between the two conditions (low ($1-3): mean_C_ = 77%, mean_U_ = 87%, *t*(22) = -1.35, *P* = 0.19; middle ($4-6): mean_C_ = 66%, mean_U_ = 45%, *t*(47) = 5.41, *P* < 0.001; high ($7-9): mean_C_ = 28%, mean_U_ = 8%, *t*(72.50) = 4.00, *P* < 0.001; Figure 2b2; see Figure 2—figure supplement 2 for rejection rates by each offer size). These results suggest that participants behaved in a strategic way to utilize their influence over the partners.”

In Methods, we also clarify how offers were generated as copied below.

Line 514: “Experimental paradigm: laboratory version. (…) In the Uncontrollable condition, participants played a typical ultimatum game: the offers were randomly drawn from a truncated Gaussian distribution (μ = $5, σ = $1.2, rounded to the nearest integer, max = $8, min = $2) on the fly using the MATLAB function ‘normrnd’ and ‘round’. Thus, participants’ behaviors had no influence on the future offers. (…)

Experimental paradigm: online version. (…) Lastly, to remove unintended inter-individuals variability in offers for the Uncontrollable condition, we pre-determined the offer amounts under Uncontrollable (offers = [$1, 1, 2, 2, 2, 3, 3, 3, 4, 4, 4, 4, 5, 5, 5, 5, 5, 5, 6, 6, 6, 6, 7, 7, 7, 8, 8, 8, 9, 9], mean = $5.0, std = $2.3, min = $1, max = $9) and randomized the order of them.”

Second, we agree with the reviewers that it is interesting to consider whether controllability could have reduced inter-individual variability in rejection rates, given that the variability in rejection rates was indeed smaller for the Controllable condition (F(47, 47) = 207.89, *P* << 0.0001). We consider this convergence in choice strategy as further proof that humans are able to exploit their environment if there is controllability, whereas being more variable under the uncontrollable condition could be a form of undirected exploration (Wilson et al., 2014), which would result in a greater inter-individual variability.

Additionally, we directly tested if our results hold even after controlling for the variability across individuals. Specifically, we ran a mixed-effect logistic regression predicting choices (accept/reject) using the offer, the condition, and the interaction terms as predictors, and individual subjects as random effects ('choice ~ 1 + offer + condition + offer*condition + (1 + offer + condition + offer*condition | subject)'). The results (Supplementary file 1b) were consistent with our originally reported results, such that there is a significant offer effect (β = 1.82, *P* <.001), no condition effect (β = -0.58, *P* = 0.57), and no interaction effect (β = -0.30, *P* = 0.10; Controllable was coded as 1 and Uncontrollable as 0). That is, choices were sensitive to offers but the sensitivity marginally reduced under the Controllable condition compared to the Uncontrollable condition. These results showed that our main results still held even after accounting for the individual variability. We have added these results and relevant discussions in supplementary information (Supplementary file 1b).

4) In the behavioral analyses, what is the rationale for grouping the offer sizes into three bins rather than using the exact levels of offer sizes? Do the key results hold if exact values are used?

We reported and displayed the binned rejection rates mainly for two reasons: (1) It had a better matched subsample size for each bin and (2) by mistake, the range of the offers were not identical between conditions for the fMRI sample ($1-9 for the Controllable condition; $2-8 for the Uncontrollable condition). We now presented the rejection rates by each offer size and added the result as Figure 2—figure supplement 2 in the supplementary information. As depicted in Figure 2—figure supplement 2, the key results hold regarding rejection rates between conditions and across offer sizes.

Furthermore, as addressed in the previous question, we added the mixed-effect logistic regression results in the paper (Supplementary file 1b). Note that this approach is statistically more stringent, and that it showed consistent results with the original binned results. Yet, we would prefer to keep the original results in the main text, because we believe the original simple analysis would help readers to gain a better and more intuitive understanding of the results.

5) It would be helpful to include an analysis of response times. Indeed, one would expect forward planning to be associated with lengthened decision times and correspondingly, for the δ parameter (or strategizing depth, or controllable condition) to be associated with longer decision times (e.g. Keramati et al., Plos Comp. Biol., 2011). Furthermore, it was recently shown that perceived task controllability increases decision times, even in the absence of forward value computations (Ligneul et al., Biorxiv). It is also good practice to include decision times as a control parametric regressor when analyzing brain activities related to a variable potentially correlated with them. Furthermore, one could expect longer reaction times for more conflicting decisions (i.e. closer valuations of reject/accept offers).

We thank the reviewers for the suggestion. We have now conducted new behavioral as well as fMRI analyses on the response times (RT) and have included these results to the supplementary information.

First, as the reviewers suggested, RT was indeed longer for the Controllable condition than the Uncontrollable condition, suggesting that controllability may involve more contemplation (Figure 2-figure supplement 3a; mean_c_ = 1.75 ± 0.38, mean_u_ = 1.53 ± 0.38; paired t-test t(47) = 4.34, *P* < 0.001). However, in either condition, neither the correlations were significant between RT and individuals’ expected influence parameter δ (Figure 2-figure supplement 3b-c), nor between RT and individuals’ self-reported controllability ratings (Figure 2-figure supplement 3d-e).

Second, following the reviewers’ suggestions, we have now conducted a new set of fMRI analyses to include trial-by-trial RT as the first parametric regressor followed by our main parametric regressor (chosen values) in the original GLM. We did not find any significant neural activation related to the RT (P*_FDR_* <.05). However, consistent with the reported result in the original submission, the vmPFC chosen value signals were still significant at *P_FDR_* <.05 and *k* > 50 after controlling for any potential RT effects (Figure 2-figure supplement 3f; peak coordinate [0, 54, -2]).

We also ran a mixed effect linear regression to examine whether or not trial-by-trial response time is correlated with the values of chosen actions as below: RT ~ 1+ condition + chosen values + condition * chosen values + (1+ chosen values | subject).

As shown in Supplementary file 1c, we found that neither the chosen value (β = 0.00, *P =* .63) nor the interaction term (β = -0.00, *P =* .43) had significant effect on RT, while the condition effect was significant (β = 0.21, *P <*.001; consistent with Figure 2-figure supplement a). This analysis suggests that in our task, RT did not have a significant relationship with chosen value, the main parametric modulator of interest. Based on these additional results, we decided to keep our original fMRI results in the main text, but to add the new results to SI (Figure 2—figure supplement 3, Supplementary file 1c).

Finally, we ran another mixed effect linear regression to examine how “conflict” ( = chosen value – unchosen value) might impact RT: RT ~ 1+ condition + conflict + condition * conflict + (1+ condition + conflict + condition * conflict | subject)

Both the conflict (β = -0.04, *P <*.005) and the condition (β = 0.13, *P <*.001) had significant impacts on RT, while there was no interaction effect (β = 0.03, *P =* .10) (Supplementary file 1d ). This result suggests that conflict did have a significant impact of on RT. We have now added this to SI (Supplementary file 1d).

6) The authors refer to the δ parameter "modeled controllability", however the model doesn't provide any account of the process of estimating controllability from observed outcomes (see Gershman and Dorfman 2019, Nature Communications or Ligneul et al., 2020, Biorxiv for examples of such models), but only reflects the impact of controllability on value computations, or the monetary amount of "expected influence" in each condition. An augmented model might include a computation of controllability, with the δ parameter controlling the extent to which estimated controllability promotes forward planning. Even if the authors don't fit such a model, they should explicitly acknowledge that their algorithm does not implement any form of controllability estimation, and might consider calling δ a "forward planning parameter". In addition, it is unclear why the authors chose to constrain the δ parameter to fluctuate between -2 and 2$ (rather than between 0 and 2$, in line with their experimental design, or with even broader bounds) and what a negative δ would imply. Also, would it make sense to exclude participants with a negative δ in addition to those with a δ greater than 2? Do all results hold under these exclusions?

Thank you and we fully agree with your suggestion. We have now revised the term “modelled controllability” to “expected influence” throughout the manuscript.

We constrained the *magnitude* of the δ to be within $2 based on the experimental design of the Controllable condition where the true δ can only be $2 or less. The first (and the last) 5 trials were excluded in model fitting, and we assumed that individuals’ δ would have been properly learned before participants could truly exploit controllability of the environment during majority of trials. Therefore, their expectation about their influence would not be completely off from the true experiences (e.g., expecting a δ of $9 after they have only seen their partners’ offer changes of $1 or $2). It would certainly be interesting in future work to adjust the design of the task, for instance with fluctuating degrees of controllability, to be able to gain more purchase on the learning itself.

The reason why we initially allowed a negative range for the δ is because in the Uncontrollable condition, due to its randomly sampled offers, subsequent offers could drop after a rejection choice and increase after an acceptance response (i.e., opposite direction from the controllable condition). For completeness and following the reviewers’ suggestion, we have now rerun the analysis without those who had a negative δ for the Controllable condition (6.3% of the fMRI sample; 5.7% of the online sample). The behavioral results still held for both the fMRI (Figure 3-figure supplement 4) and the online samples (Figure 4-figure supplement 3). Specifically, there were statistically significant differences between the two conditions in (i) the offer size (fMRI sample: *t*(44) = 5.05, *P <*.001; online sample: *t*(1,265) = 22.94, *P <*.001), (ii) the rejection rates for the middle (fMRI sample: *t*(44) = 5.33, *P <*.001; online sample: *t*(1,265) = 10.23, *P <*.001) and high offers (fMRI sample: *t*(38) = 4.68, *P <*.001; online sample: *t*(934) = 31.40, *P <*.001), (iii) the perceived controllability (fMRI sample: *t*(36) = 3.67, *P <*.001; online sample: *t*(1,265) = 26.23, *P <*.001), and (iv) the δ (fMRI sample: *t*(44) = 5.14, *P <*.001; online sample: *t*(1,265) = 19.54, *P <*.001). In addition, (v) the δ was positively correlated between the two conditions (fMRI sample *r* = .40, *P* <.01; online sample: *r* = .25, *P* <.001), and (vi) the δ and the mean offers were positively correlated (fMRI sample *r* = .86, *P* <.001; online sample: *r* = .71, *P* <.001). We now provide this set of analyses and results in SI (Figure 3—figure supplement 4 and Figure 4—figure supplement 3).

7) While the authors performed a parameter recovery analysis, they did not report cross-parameter correlations, which are important for interpreting the best-fitting parameters in each condition. Furthermore, it is good practice to perform model recovery analyses on top of parameter recovery analyses (Wilson and Collins, 2019, eLife; Palminteri et al., 2017, TiCS) in order to make sure that the task can actually distinguish the models included in the model comparison. As a result, the conclusions based on model comparison and parameters values (that is, a significant part of the empirical results) are uncertain. The cross-correlation between parameters and model recovery analysis should be reported as a confusion matrix.

Thank you and we fully agree with your suggestion. We have now added the cross-parameter correlations as well as the model recovery results as confusion matrices in the paper.

Figure 4—figure supplement 2 illustrates the cross-parameter correlations. We did not find any strong correlations (*r* > 0.5) between parameters. The α (sensitivity to norm prediction error) and the F0 (initial norm) were moderately correlated (*r* = -0.39) under the Controllable condition for the fMRI sample. However, these parameters were still independently identifiable (parameter recovery α: *r* = 0.57, *P <* 0.001, F0: *r* = 0.66, *P <* 0.001; please see Figure 3—figure supplement 3b-c).

We have now added model recovery results in the supplementary information (added as Figure 3—figure supplement 1). To examine whether our task design is sensitive enough to distinguish the models, we simulated each model where we fixed the inverse temperature at 10 and constrained the δ to be positive (between [0 2], inclusive) to make it similar to the actual empirical behavior we found. Other parameters were randomly sampled within the originally assumed range. We ran 100 iterations of simulation where in each iteration, behavioral choices of 48 individuals (which is equal to our fMRI sample size) were simulated. Next, we fit each model to each model’s simulated data where all the settings were identical to the original settings. Consistent with our original method, we calculated average DIC scores and determined the winning model of each iteration. In this set of new analysis, we focused on three distinct types of models in our model space, namely, model-free (MF), no FT (0-step), and FT (2-step) models. Note that we chose the 2-step FT model as a representative of the FT models due to both its simplicity (i.e., per Occam’s razor) and due to the similarity amongst the FT models. This additional model recovery analysis suggests that our task can actually distinguish the models.

8) The parameters of the adaptive social norm model exhibit fairly poor recoverability, particularly in the controllable condition. The motivation for using this model is that it provided the best fit to subjects data in a prior uncontrollable ultimatum game task, but perhaps such adaptive judgment is not capturing choice behavior well here. It would be helpful to see a comparison of this model with one that has a static parameter capturing each individual's subjective inequity norm.

Thank you for this insightful suggestion. Following your suggestion, we have now examined an alternative 2-step model that has a static norm throughout all trials within each condition. The DIC scores of the alternative model (“static”) were higher than the original 2-step FT model using Rescorla-Wagner learning (“RW”) in both conditions (smaller DIC score indicates a better model fit; added as Figure 3—figure supplement 2). This new analysis suggests that the adaptive norm model still offers a better account for behavior than the static norm model.

9) The authors stated that future actions are deterministic (line 576) contingent on the utility following the immediate reward. If so, is Figure 3a still valid? If all future actions are deterministic, there should be only one path from the current to the future, rather than a tree-like trajectory.

Thank you for bringing up this point. Figure 3a was indeed erroneous and we have now revised it by highlighting one path to represent our model better.

10) The MF model, and the rationale for its inclusion in the set of models compared, needs to be explained more clearly. The MF model appears to include no intercept to define a base probability of accepting versus rejecting offers, which makes it hard to compare with the other models in which the initial norm parameter may mimic such an intercept.

We called it the MF model because it updates Q-values in light of the cached values of the following state (as opposed to the 0-step model, which only considers the current state), yet without a state transition component (as opposed to the FT models that reflect state transitions, which we conceptualized as controllability). All other components including the utility function of the immediate rewards, and the variable initial norm and norm learning incorporated in the utility function are shared across all the candidate models. In common with the other candidate models, the MF model also includes an initial norm parameter, which captures the base probability of accepting versus rejecting offers. We have now revised the manuscript to clarify the possible confusion.

Line 181: “We compared models that considered from one to four steps further in the future in addition to standalone social learning (‘0-step’) and model-free, non-social reinforcement learning (‘MF’). The 0-step model only considers the utility at the current state. The MF model updates and caches the choice values that reflect the following states, yet without a state transition component (as opposed to the FT models that reflect state transitions, which we conceptualized as controllability). All other components including the utility function of the immediate rewards, and the variable initial norm and norm learning incorporated in the utility function are shared across all the candidate models.”

11) The fact that the vmPFC encoded total future + current value (2-step) and not current value (0-step) suggests that it might be specifically involved in computing future values but the authors do not report directly the relationship between its activity and future values. How correlated are the values from the 0-step model and the 2-step model? And more importantly, if vmPFC is associated with TOTAL value but not the CURRENT value, should that mean the vmPFC is associated with the FUTURE value only? It might make more sense to decompose the current value and future value both from the winning 2-step model, and construct them into the same GLM without orthogonalization.

Thank you for this comment. We originally considered the model-based fMRI analysis to be a biological validation of the models more than a way to delineate the different neural substrates of current and future values. That is, the lack of significant neural signals tracking values from the 0-step model may suggest that the 0-step model is less plausible than the 2-step model at the neurobiological level, despite the fact that value estimates are highly correlated between the two models (mean correlation coefficient was 0.74 for the Controllable condition and 0.84 for the Uncontrollable condition). Nevertheless, we agree that it would be interesting to examine if the current and future value terms under the same (2-step FT) model might be encoded by different neural substrates. Thus, we have now run a new set of GLMs with both the current and the future values without orthogonalization. We found that the current value-alone signal was encoded in the vmPFC (peak voxel [2, 52, -4]) and the dmPFC ([2, 50, 18]), and the future value-alone signal was tracked by the right anterior insula ([34, 22, -12]), at the threshold of *P* <.001, uncorrected. Although these results did not survive the more stringent threshold applied to the main results (*P_FDR_* <.05, *k* > 50), all survived the small volume correction at *P_SVC_* <.05. Figure 5-figure supplement 2 was displayed at *P* <.005, uncorrected, *k* > 15. Together with our main result, these results indicate that the vmPFC encodes both current and total values estimated from the 2-step FT model; and that current and future value signals also had distinct neural substrates (dmPFC and insula). We have now added this new analysis in SI (Figure 5—figure supplement 2).

12) The vmPFC result contrast averages across the controllable and uncontrollable conditions (line 629). Why did the authors do so? Wouldn't it be better to see whether the "total value" is represented differently between the two conditions.

We apologize for not being clear about this point in the previous version of the paper. We showed the averages because there was no significant difference between the two conditions both in the whole brain analysis (*P_FDR_* <.05) as well as in the ROI analyses (Figure 5c). This result might seem puzzling at a first glance; however, it is consistent with our computational modelling results in the sense that people simulated 2 steps regardless of the actual controllability of the environment and that they needed to engage the vmPFC to do so. We have now added more clarification and discussion on this finding in the revised manuscript.

Line 325: “These analyses showed that the BOLD signals in the vmPFC tracked the value estimates drawn from the 2-step planning model across both conditions (*P_FDR_* < 0.05, *k* > 50; Figure 5a, Supplementary file 1e), and there was no significant difference between the two conditions (*P_FDR_* < 0.05). In contrast, BOLD responses in the vmPFC did not track the trial-by-trial value estimates from the 0-step model, even at a more liberal threshold (*P* < 0.005 uncorrected, *k* > 50; Figure 5b, Supplementary file 1f). (…) These findings suggest that individuals engaged the vmPFC to compute the projected total (current and future) values of their choices during forward thinking. Furthermore, vmPFC signals were comparable between the two conditions both in the whole brain analysis and the ROI analyses. Consistent with our behavioral modeling results, these neural results further support the notion that humans computed summed choice values regardless of the actual controllability of the social environment.”

Line 419: “In addition, we did not find any significant differences in neural value encoding between the conditions. These results suggest that participants still expected some level of influence (controllability) over their partners even when environment was in fact uncontrollable. Furthermore, δ was positively correlated between the conditions, indicating the stability of the mentally simulated controllability across situations within an individual. We speculate that people still attempted to simulate future interactions in uncontrollable situations due to their preference and tendency to control (Leotti and Delgado, 2014; Shenhav et al., 2016).”

13) The analysis of the relation between the vmPFC β weights and the difference between self-reported controllability beliefs and model-derived controllability estimates (Figure 5 d and e) is not adequately previewed. The hypothesis for why vmPFC activity might track this metric is unclear. Moreover, the relation between the two in the uncontrollable condition is somewhat weak. The authors should report the relation between vmPFC β weights and each component of the difference score (modeled and self-report controllability), and clearly motivating their intuition for why vmPFC activation might be related to that metric. If the authors feel strongly that this analysis is important to include, it would be meaningful to see whether the brain data could help explain behavioral data. For example, a simple GLM could serve this purpose: mean_offer ~ β(vmPFC) + self-report_controllability + model_controllabilty. Note that the authors need to state the exploratory nature if they decide to run this type of analysis.

Thank you for this comment. This was indeed an exploratory analysis. Thus, we now have clarified this in the revised manuscript and moved relevant results to SI. We sought to explore the neural correlates of the belief-behavior disconnection because we saw a discrepancy between the self-reported belief and the δ parameter particularly for the Controllable condition (*r* = .004, *P =* .98; although a moderate correlation exists for the Uncontrollable condition: *r* = -.43, *P <*.01) which we had not expected, and seemed rather interesting. We therefore explored the relationship between individuals’ neural sensitivity to total decision values in the vmPFC and each of the two measures, and examined whether the vmPFC sensitivity mediates the relationship between the measures. However, the vmPFC ROI coefficients were not correlated with each separate component, either the perceived controllability (Controllable condition: *r* = .05, *P =* .76; Uncontrollable condition: *r* = .21, *P =* .18) or the δ (Controllable condition: *r* = .23, *P =* .11; Uncontrollable condition: *r* = -.27, *P =* .07). To clarify the explorative nature of our analysis, we edited corresponding sections as follows.

Line 351: “Furthermore, in an exploratory analysis, we examined the behavioral relevance of these neural signals in the vmPFC beyond the tracking of trial-by-trial values. [...] These results suggest that the meaning of vmPFC encoding of value signals could be context dependent – and that heightened vmPFC signaling in uncontrollable situations is related to overly optimistic beliefs about controllability.”

Furthermore, we conducted the linear regression suggested by the reviewers. The results showed that only the δ, not the vmPFC ROI coefficient or perceived controllability predicted the mean offer size (Author response table 1). However, this regression does not examine the relationship between neural sensitivity and the belief-behavior disconnection, and thus we did not add the result to the revised manuscript.

**Author response table 1. sa2table1:** 

	Estimate	SE	t	p
Intercept	2.87	0.61	4.70	0.00
vmPFC	-0.26	0.16	-1.60	0.12
PC	0.01	0.01	1.51	0.14
Δ	1.76	0.21	8.34	0.00

14) The authors might also report the neural correlates of the internal norm and the norm prediction error (line 544). If the participants indeed acquired the social controllability through learning, they might form different internal norms in the two conditions, hence the norm prediction error might also differ.

Thank you for this excellent prompt. We have now conducted a new set of whole brain analyses to examine the neural correlates of norm prediction errors and internal norms by entering them as parametric modulators in two separate GLMs.

The norm prediction error signals were found in the ventral striatum (VS; [4, 14, -14]) and the right anterior insula ([32, 16, -14]) for the Controllable condition, and in the anterior cingulate cortex ([2, 46, 16]) for the Uncontrollable condition at *P_FWE_* <.05, small volume corrected. These regions have been suggested to encode the prediction errors in the similar norm learning context (Xiang et al., 2013). Next, we contrasted the two conditions and found that the ventral striatum ([4, 14, -14]) and the right anterior insula ([32, 16, -14]) activations were significantly greater for the Controllable condition than for the Uncontrollable condition (*P_FWE_* <.05, small volume corrected) whereas the ACC ([2, 46, 16]) activation under the Uncontrollable condition was not significantly greater than the Controllable condition at the same threshold. Figure 5—figure supplement 3 was displayed at *P* <.05, uncorrected, *k* > 120.

The norm-related BOLD signals were found in the ventral striatum ([10, 16, -2]) for the Controllable condition, and in the right anterior insula ([28, 16, -6]) and the amygdala ([18, -6, -8]) for the Uncontrollable condition at *P_FWE_* <.05, small volume corrected. However, the whole brain contrast showed no difference between the conditions. Figure 5—figure supplement 4 was displayed at *P* <.01, uncorrected, *k* > 50.

(15) Specific aspects of the experimental design may have influenced the observed results in ways that were not controlled. For example, it is not only the magnitude and controllability of outcomes that differed between the controllable and uncontrollable conditions, but also the uncertainty. It is possible that the less variable offers encountered in the controllable condition may have driven some of the results. The authors should acknowledge that the possible role of autocorrelation and uncertainty on behavioral and modeling results.

Thank you for bringing up this point. Autocorrelation and uncertainty are indeed inherent features of having control. To address this concern, we examined the relationship between uncertainty/autocorrelation and our key outcome variables. To this end, we first operationalized uncertainty as the standard deviation of offers within each condition (“offer SD”). We entered the offer SD with the condition variable in a regression predicting the δ parameter (Δ ~ 1 + offer SD + condition) and the self-reported perceived controllability (perceived control ~ 1 + offer SD + condition). The results showed that the condition still predicted the δ (β = 0.36, *P* <.05; Supplementary file 1g a) whereas the offer SD had no significant impact on the δ (β = -0.03, *P* = 0.92; Supplementary file 1g a). Similarly, for the self-reported controllability, the condition (β = 21.09, *P* <.001; Supplementary file 1g b) had a significant effect whereas the offer SD did not (β = 15.60, *P* = .16; Supplementary file 1g b).

Next, to examine the autocorrelation issue, we computed the sample autocorrelation of offers using ‘autocorr’ function in MATLAB. Indeed, 30 out of 48 subjects showed significant autocorrelation at lag 1 (“ACF1”) for the Controllable condition whereas none had significant autocorrelation for the Uncontrollable condition. Next, we entered ACF1 into a regression model similar to the one stated above. We found that for δ, the condition effect became marginal (β = 0.46, *P* = .07; Supplementary file 1g c), and there still was no significant ACF1 effect (β = -0.21, *P* = .59; Supplementary file 1g c). For the perceived controllability, the condition effect was marginal (β = 18.14, *P* = .05; Supplementary file 1g d) but the ACF1 effect was not significant (β = 7.93, *P* = .58; Supplementary file 1g d). We have added these results in the supplementary information (Supplementary file 1g).

In summary, we did not find evidence that uncertainty or autocorrelation had a major impact on our controllability measures. Yet, controlling for the autocorrelation weakened the impact of controllability condition. We added this as a limitation of the study in Discussion.

Line 466: “In addition, a task that carefully controls for uncertainty and autocorrelation confounds would help better understanding the accumulative effects of social controllability. Although we did not find evidence that uncertainty or autocorrelation affected the expected influence or self-reported controllability, we found that the impact of the condition on the expected influence and the self-reported perceived controllability became marginal when controlling for the autocorrelation (*P* = 0.07 for the expected influence; for self-reported controllability *P* = 0.05) (Supplementary file 1g).”

(16) Moreover, asking participants to repeatedly rate their perception of controllability almost certainly influenced and exacerbated the impact of this factor on choices. It would have been very useful to perform a complementary online study excluding these ratings to ensure that controllability-dependent effects are still evident in such a case.

Sorry for the lack of clarify in our original writing. In fact, participants were not repeatedly asked to rate the perception of controllability within the trials. This was on purpose because we shared the same concern you raised here. As such, we intentionally avoided such issue by only asking participants to rate their perception of controllability *at the end of the task*. We have now clarified this point in the revised manuscript.

Line 98: “At the end of the task, after all the trials were completed, participants rated how much control they believed they had over their partners’ offers in each condition using a 0-100 scale (‘self-reported controllability’ hereafter). In the fMRI study, on 60% of the trials, participants were asked about their emotional state (“How do you feel?”) on a scale of 0 (unhappy) to 100 (happy) after they made a choice (i.e., 24 ratings per condition; see Figure 1—figure supplement 1).”

[Editors' note: further revisions were suggested prior to acceptance, as described below.]A primary concern is that the manuscript does not provide sufficiently strong support for the claim that the vmPFC supports forward planning, particularly in light of the new neuroimaging analyses performed as part of this revision. Reviewer 3 has a concrete suggestion for how this claim might be strengthened with a model comparison analysis. If further evidence for the claim is not found/provided, it should be tempered. Reviewer 2 also questions whether it is useful and sensible to retain the MF model in the set of compared models, and both reviewers note a few areas where clarification, greater methodological detail, or further interpretation are warranted.Please carefully consider each of the reviewers suggestions as you revise your manuscript.

Thank you and all reviewers for your thoughtful comments. We are thrilled that all reviewers find the revised manuscript significantly improved. We are especially grateful for these two final suggestions made by reviewers. Following Reviewer 3’s suggestion, we have now implemented a neural model comparison using the recommended MACS toolbox; this analysis further substantiated our finding related to the vmPFC and relevant methods/results have been added to the revised manuscript. Following Reviewer 2’s suggestion, we have removed the MF model from the main text. Finally, we have also made every effort to address all remaining concerns. Please see our point-by-point response below.

Reviewer #2:The authors have revised their manuscript considerably and addressed a number of concerns raised in the initial review, with their additional analyses and detailed clarification. I particularly appreciate that the authors took the courage to dive into the direct comparison of findings between the social and non-social groups, which provided new insights. Furthermore, the revised Introduction is more thought-provoking with relevant literature included. Now the conclusions are better supported as it stands, and these findings are certainly going to be exciting additions to the literature of social decision neuroscience.Here I have a few additional points, more for clarification.(1) In response to comment #2, the authors might unpack the significant interaction result, to explicitly show "that the non-social context reduced the impact of nPE on emotional feelings." Also in the same LME model, I am curious about the significant "Controllable × social task (***)" interaction (β = -5.06). Does this mean, being in the Controllable + Social group, the emotion rating is lower? How would the authors interpret this finding?

Thank you for this suggestion. To unpack the interaction effect of norm prediction error and task type, we conducted residual correlations and plotted the mean coefficient in Figure 2-figure supplement 1h. Specifically, we used the regression coefficients from the original mixed-effect regression ('emotion rating ~ offer + norm prediction error + condition + task + task*(offer + norm prediction error + condition) + (1 + offer + norm prediction error | subject)') and calculated the residual, which should be explained by the differential impact of nPE between social and non-social tasks. Correlation coefficients between the residuals and nPE were plotted for each task condition. Note that the non-social group was coded as the reference group (0 for the group identifier) in the regression. We also added this figure in our manuscript as Figure 2—figure supplement 1h. This result indicates that the impact of nPE on emotion was stronger in the Social than in the non-social Computer task, revealing an interesting joint contribution of PE and the social context to subjective states. We speculate that this is due to the interpersonal nature of the social version of the game and suggest that the cause of this effect deserves further instigation in future studies.

Regarding the Controllable × social task interaction, we applied a similar residual approach and found that emotion rating was lower in the Social and Controllable condition compared to other condition × task type combinations (Figure 2-figure supplement 1i ). We speculate that exerting control over other people – compared to not needing to exert control over other people or playing with computer partners – might be more effortful (as shown by our RT results). Intentionally decreasing other people’s portion of money might also induce a sense of guilt. We have now added this figure in Figure 2—figure supplement 1i.

(2) In response to comment #5 regarding response time with the additional LME analyses, I wonder which distribution function was used? We know that RT data is commonly positively skewed, so a log-normal or a shifted log-normal should be more accurate.

Thank you for your suggestion. Our previous results were based on normal distribution. Following the reviewer’s suggestion, we have now re-ran the regressions using log-normal function. New results related to “chosen value” were similar to the previous version (Author response table 2) in that only the condition effect was significant (Author response table 3) .

**Author response table 3. sa2table3:** 

Name	Estimate	SE	t	DF	p-value
Intercept	1.60	0.06	28.72	2860	0.000
Condition (***)	0.13	0.03	4.39	2860	0.000
Conflict (**)	-0.04	0.01	-2.98	2860	0.003
Condition × conflict	0.03	0.02	1.66	2860	0.096

**Author response table 2. sa2table2:** 

Name	Estimate	SE	t	DF	p-value
Intercept	1.52	0.06	24.43	2860	0.000
Condition (***)	0.21	0.06	3.71	2860	0.000
Chosen value	0.00	0.01	0.48	2860	0.630
Condition × chosen value	0.00	0.01	-0.80	2860	0.426

We ran a mixed effect linear model (RT ~ 1+ condition + chosen values + condition * chosen values + (1+ chosen values | subject)) to test whether chosen values predict response times. We found that neither the chosen value coefficient (β = 0.00, *P* = .63) nor the interaction term (β = -0.00, *P* = .43) was significant, while the condition effect was significant (β = 0.21, p <.001; consistent with Figure 2—figure supplement 3a). *** *P* < 0.001

We ran a mixed effect linear model (RT ~ 1+ condition + conflict + condition * conflict + (1+ conflict | subject)) to test whether conflicts (values of the chosen action – values of the unchosen action) affect response times. Both the conflict (β = -0.04, p <.005) and the condition (β = 0.13, p <.001) had significant impacts, while there was no interaction effect (β = 0.03, p = .10), suggesting that conflict did have a significant impact of on RT. ** *P* < 0.01; *** *P* < 001.

The new regression analysis examining “conflict” also showed similar results to the previous version (Author response table 3) except that the condition x conflict interaction effect became significant (Supplementary file 1d) from being marginal in the previous version. We replaced the previous results with these new results.

(3) I retain my initial comment regarding the inclusion of the MF model. The task is deterministic – participants get what appears if they accept and 0 if reject. In fact, the model is making a completely different prediction: according to the Q-value update, if the participant chose an "accept" and then indeed received a reward, then they should repeat "accept". But in the current task design, such a "positive feedback" would make the participants feel they are perhaps too easy to play with, and will be more likely to choose "reject" on the next trial. In essence, the MF model is not even capturing the behavioral pattern of the task, hence it does not seem to be a good baseline model. Rather, the 0-step model is okay enough to be the reference model.

Thank you for the suggestion. We agree that the MF model clearly did not capture real subjects’ behaviors during this social interaction game and have now removed the MF results from the main text. For completeness, we kept MF-relevant descriptions and results in Figure 3—figure supplement 5 for readers who might be interested.

Reviewer #3:The authors have made very significant efforts to respond to a diversity of concerns and to amend their paper accordingly. The revised version is thus more complete and I believe that the main argument of the paper has been made stronger.In many cases, the authors have appropriately adjusted their language in order to better align their conclusions with the data (e.g. renaming the δ parameter expected influence parameter) and I think that this paper can constitute an interesting addition to the field.However, I am still slightly skeptical about the reach of neuroimaging results and I believe that some limitations of the paradigm may be more explicitly discussed.A. Neuroimaging.The authors have performed valuable additional analyses regarding the norm and norm prediction errors signals which can be of interest for the field. But I believe that our main concerns about vmPFC effects have not been fully addressed. Indeed, the authors still write that the vmPFC constructs "the total values (both current and future) of current actions as humans engaged in forward planning during social exchange". However, when splitting the analysis of current and future values, the encoding of future values was found in the insula whereas the vmPFC only encoded current values. The authors claim that the lack of encoding of total values derived from the 0-step FT model constitutes evidence in favor of forward planning, but it could be that this lack of evidence is driven by a poorer fit of current (rather than total) values by this simpler model. In order to better substantiate their claim about vmPFC's role, the authors may want to perform a model comparison at the neural level by comparing GLMs (using for example the MACS toolbox) including current value only, current value and future value, future value only or total value. Alternatively, they could analyze the first-level residuals produced by GLMs including alternatively current value, future value and total value (all based on FT-2). If their interpretation is correct, GLMs equipped with a parametric regressor for total value should be associated with smaller residuals in the vmPFC.

Thank you for this insightful comment, which has further helped strengthen the paper. Following your suggestion, we performed a model comparison at the neural level using the MACS toolbox. We compared four different GLMs: (i) the GLM with total value (TV) as one regressor (our original GLM), (ii) the GLM with current and future value (CVandFV) as two regressors (without orthogonalization), (iii) the GLM with only current value (CV), and (iv) the GLM with only future value (FV). All value regressors were estimated from the 2-step forward thinking (FT) model. We found that our original GLM with total value had clearly higher exceedance probability in the vmPFC compared to other candidate GLMs. We added these results in our manuscript as Figure 5—figure supplement 3 and briefly discussed it the Results section.

Line 333 (line numbers are based on the clean version): We also conducted model comparison at the neural level using the MACS toolbox (see Figure 5—figure supplement 3 for details) and found that the vmPFC encoded total values rather than only current or future values.

Regarding the behavior-belief disconnection analysis, I think that it would be more sensical to study the ratio rather than the difference between behavior and subjective reports, since these two measures are qualitatively different.

Thank you for the suggestion. To clarify, both measures were standardized before we calculated the distance. Our measure is simply the signed distance between the standardized belief and the standardized δ, or the signed and scaled “disconnect distance” as illustrated in Author response image 1 (our measure = √(2* [disconnect distance]^2^)). This approach has been commonly used in the cognitive neuroscience literature (Carter et al., 2012; Chung et al., 2015; Zhu et al., 2014).

**Author response image 1. sa2fig1:** 

We believe that such an Euclidean distance is a better way to operationalize the “disconnect” than a ratio, because a ratio leads to uneven weights. For example, we intend to quantify the disconnections between [behavior: 1 vs belief: 2] and [behavior: 2 vs belief: 1] to have equal magnitude, but opposite signs. The suggested ratio measures generate different magnitudes (1/2=0.5 and 2/1=2), whereas the Euclidean distance captures the difference in signs with keeping the magnitude of distances the same as we intended (1-2=-1 and 2-1=1).

Finally, it might be worth providing the reader with a brief discussion of the other neural substrates uncovered by the most recent analyses (dmPFC, insula, striatum, etc.).

Thank you for the suggestion. We discussed the findings in the result section.

Line 356: “In addition, we examined whether norm prediction errors (nPEs) and norm estimates themselves from the 2-step FT model were tracked in the brain. We found that nPEs were encoded in the ventral striatum (VS; [4, 14, -14]) and the right anterior insula (rAI; [32, 16, -14]) for the Controllable condition (Figure 5—figure supplement 4a), while these signals were found in the anterior cingulate cortex (ACC, [2, 46, 16]) for the Uncontrollable condition (Figure 5—figure supplement 4b) at *P_FWE_* <.05, small volume corrected. […] Taken together, these results suggest that the controllability level of the social interaction modulates neural encoding of internal norm representation and adaptation, expanding our previous knowledge about the computational mechanisms of norm learning (Gu et al., 2015; Xiang et al., 2013).”

B. Behavioral paradigm.I believe that the authors should provide a few more details in the methods and acknowledge a few limitations in their discussion.First, unless I am mistaking the method used to decide on block order (i.e. C or U first) was not reported. Was the "illusion of control" in the uncontrollable condition driven by the subset of participants who passed the controllable block first? If this is the case, then it might add some plausibility to the interpretation of subjective controllability ratings in the uncontrollable condition as an "illusion of control" (persistence of a control prior). In other words, I think that the authors should refrain from interpreting the raw value of these ratings as an illusion of control (perhaps not all participants understood the meaning of the rating, perhaps they were too lazy to move the cursor until 0, etc.).

The condition order was counterbalanced (see manuscript line 76 and 875). Here, we also address the two different aspects of your comment below.

1) Regarding examining a potential order effect and proposed potential “Bayesian”-like explanation for the reported behavior: This is an interesting and appealing hypothesis that we have now examined. However, we did not identify any order effect in terms of the expected influence or the self-reported controllability (Supplementary file 1h), suggesting that there was no evidence that “illusion of control” was induced by completing the controllable block first. Priors of controllability might still exist in the subjects – but based on our data, such prior would be more likely due to pre-existing individual differences (e.g. childhood experience in environmental controllability) than task-induced priors per se. Although we did not survey these individual differences, future studies could systematically investigate this interesting question. We have added this to our discussion.

Although we did not find any order effect on the expected influence parameter or self-reported belief, future studies would be needed to probe task-induced priors more thoroughly. PC represents self-reported perceived controllability. C represents the Controllable condition and U represents the Uncontrollable condition.

2) Regarding potentially noisy subject reports: Indeed, whenever we include subjects’ self-reports in empirical studies, there is always the risk that these reports are potentially noisy and reflect non-task related factors (e.g., too lazy to move the cursor). In fact, this has been a longstanding topic for discussion in the history of psychology and cognitive neuroscience (e.g. see Stone, Arthur A., et al., The science of self-report: Implications for research and practice. 1999 for an overview). However, most researchers agree that subjective data are still valuable as they offer unique insights – and perhaps the most direct insight – into one’s subjective world, which is also specific to human studies. Based on our results, we suggest that both measures (objective and subjective) contribute to the illusion of control. For the purpose of this manuscript, we have clarified these points throughout the text and tuned down any direct claim based on the sole association between self-reports and illusion of control.

While it does not necessarily implies an illusion of control, the fact that participants still relied on on forward planning in the uncontrollable condition (as indexed by the expected value parameter) is presumably what prevented authors to really isolate the neural substrates of strategic controllability-dependent forward planning, and it might thus be mentioned as a limitation of the paradigm.

Thank you for this comment. We added this limitation in the Discussion section.

Line 501: “Second, the lack of clear instruction in different controllability conditions in our study may have affected the extent to which individuals exploit controllability and develop illusion of control. Future studies implementing explicit instructions might be better suited to examine controllability-specific behaviors and neural substrates.”

I believe that it is also important to mention explicitly the fact that a third and a quarter of the data was excluded from the analyses of behavioral and fMRI data (i.e. first and last five trials of each block) respectively and the rationale for this exclusion may be discussed.

Thank you for this comment. We added the total number of the trials in the paragraph so that the proportion to be excluded is clearer.

Line 193: “In model fitting, we excluded the first five trials out of 40 trials for the fMRI sample (30 trials for the online sample) to exclude initial exploratory behaviors and to focus on stable estimation of controllability. We also excluded the last five trials because subjects might adopt a different strategy towards the end of the interaction (e.g. “cashing out” instead of trying to raise the offers higher).”

The authors wrote that "a task that carefully controls for uncertainty and autocorrelation confounds would help better understanding the accumulative effects of social controllability", which is a good start, but it would be in my opinion important to explicitly acknowledge that change in controllability were confounded with change in uncertainty about upcoming offers.

Thank you for this comment. We revised this part in the discussion.

Line 496: **“**We did not find evidence that uncertainty or autocorrelation affected the expected influence or self-reported controllability and that reduction in uncertainty might be an inherent feature to controllability (Supplementary file 1g). Still, future experimental designs which dissociate change in uncertainty from change in controllability may better address potentially different effects of controllability and uncertainty on choice behavior and neural responses.”

I would be curious to hear the authors' insight about why participants in the online study (and to some extent in the lab) accepted more often the low offers in the controllable condition. It seems somehow counterintuitive and could mean that participant behaved in a more "automatic" and perseverative way in the controllable condition.Related to this last point, is it possible that the δ parameter (or expected influence) simply captures a perseverative tendency in rejection/acceptance of offers? This might explain the disconnection between behavior and belief, as well as the positive value of this parameter in the uncontrollable condition, correlated to that of the controllable one. That perseveration increases in the controllable condition would be logical (since that condition allows participants to reach their goal by doing so) and it would therefore still be of interest in the context of this social controllability study. Perhaps the authors could exclude this possibility by running adding a perseveration mechanism to their model, as it is often done in the RL literature?

Thank you for this comment. First, regarding why online participants seemed to accept more often in the low offers in the controllable condition, we would like to point out that certain type of subjects was over-represented in the low offer bin – those who have low expected influence (δ) and thus ended up with more low offers (yet still more likely to accept these low offers). Individuals who had high δ were much less likely – sometimes never – to reach the lowest offers and were thus under-represented in the “low offer” bin in the Controllable condition. In contrast to the controllable condition, everyone experienced the same number of low offers in the Uncontrollable condition. Thus, the over-representation of “low δ” and under-representation of “high δ” subjects is the main reason why low offers are more accepted in the Controllable condition.

Second, both our previous RT analysis and new analysis of shift ratio provide evidence that perseverance is likely to be less, rather than more, prevalent, in the Controllable condition. Habitual behaviors and behavioral perseverance are generally associated with faster time to respond (e.g., see Hardwick et al., "Time-dependent competition between goal-directed and habitual response preparation." *Nature human behaviour* 3.12 (2019): 1252-1262; Keramati, Dezfouli, and Piray. "Speed/accuracy trade-off between the habitual and the goal-directed processes." *PLoS computational biology* 7.5 (2011): e1002055). Here, we found that human subjects showed longer, instead of shorter RT, in the Controllable condition (Figure 2d), suggesting that they were likely to engage more deliberation to exploit controllability.

We also conducted new analysis to examine shift ratio (i.e., the number of the trials where the choice was shifted from the previous trial divided by the total number of the trials). We found that shift ratio was higher, rather than lower, for the Controllable condition than the Uncontrollable condition (mean_C_ = 52.5%, mean_U_ = 36.2%, *t*(47) = 6.62, *P* <.001), and the shift ratio was not correlated between the two conditions (*R* = .24, *P* = .10). Together with the RT analysis, these results suggest that subjects were less likely to be habitual in the Controllable condition. We have now added the new shift ratio analysis in Figure 2—figure supplement 4.